# FAFO: Lossy KV Cache Compression for Lossless Inference Acceleration via Draftless Fumble Decoding

Hoang Anh Duy Le [* 1]   Shaochen (Henry) Zhong [* 1]   Yifan Lu [1]   Yingtong Dou [2]   Jiayi Yuan [1]
Yu-Neng Chuang [1]   Xiran Fan [2]   Guanchu Wang [1]   Yuzhong Chen [2]   Xia (Ben) Hu [1]

## Abstract

*Lossy KV cache compression* is a well-explored subfield of machine learning efficiency, with improved latency being one of its major gains. However, lossy compression techniques can fumble from time to time, exhibiting various, and often catastrophic, failure patterns that are not only difficult to resolve but sometimes even hard to identify, making direct deployment of models with compressed KV cache a risky endeavor. In this work, we explore a way to preserve *lossless* generation quality while still benefiting from the acceleration provided by KV cache compression. Specifically, we draw inspiration from the *n-gram candidate pool decoding* paradigm where we purposely allow the model to Fumble Around with compressed KV cache to generate multiple lossy "n-gram guesses", while in parallel Find Out via lossless verification in the same forward pass. From a conceptual standpoint, our proposed framework is compatible with all typical static or dynamic KV cache compression methods from the token dropping realm, thus opening up a new avenue for the stagnant n-gram decoding paradigm. Practically, we show that this framework presents many useful traits that similar draftless baselines (e.g., Self-Speculative Decoding) cannot achieve, such as requiring only one set of KV cache and being far less sensitive to model, task, and input-length scenarios. Our comprehensive empirical results show FAFO provides 1.20-2.71$\times$ latency speedup over the original model, while consistently outperforming other lossless + draftless solutions.

## 1. Introduction and Background

Transformer-based Large Language Models (LLMs) have demonstrated strong capabilities across a wide range of general and specialized tasks. However, one innate challenge of relying on transformer-based architectures is the Key-Value (KV) cache, which is necessary for efficient inference. The KV cache typically grows linearly with batch size and sequence length, and its sheer size creates a significant efficiency bottleneck for model-serving systems, as noted in prior art like Pope et al. (2023); Fu (2024). For these reasons, *KV Cache Compression* as a subfield has received major interest and advancement over the past year (Yuan et al., 2024; Luohe et al., 2024), with many lossy efficiency methods proposed to allow models to process using only compressed KV cache, thereby achieving significant savings on memory footprint and improved latency.

In this work, we present **a framework that maintains lossless generation quality while using lossy KV cache compression as its means** — Fumble Around and Find Out (FAFO) — within the draftless efficient decoding paradigm. Specifically, FAFO leverages a modified version of the n-gram candidate-pool decoding paradigm pioneered by Lookahead Decoding (Fu et al., 2024) and introduces several significant advancements. Empirically, FAFO yields a 1.20-2.71$\times$ latency speedup over vanilla decoding while preserving lossless generation quality, outperforming other lossless, draftless solutions by a large margin and proving particularly effective on repetition-heavy tasks such as summarization, writing assistance, and coding.

### 1.1. Lossy KV Cache Compression Can Sometimes Fail, and Fail so in Unpredictable Ways

One unresolvable challenge of lossy efficiency approaches is that they do fail under certain combinations of model, method, task, setting, compression rate, etc. Prior benchmark works like Yuan et al. (2024) indicate that when they fail, they often fail catastrophically.[1] Such failure-triggering

---

[*]Equal contribution  [1]Department of Computer Science, Rice University  [2]Visa Research. Correspondence to: Hoang Anh Duy Le <el72@rice.edu>.

*Proceedings of the 43$^{rd}$ International Conference on Machine Learning*, Seoul, South Korea. PMLR 306, 2026. Copyright 2026 by the author(s).

---

[1]See examples in Appendix B about the recorded failure modes of established methods like H2O (Zhang et al., 2023b), SnapKV (Li et al., 2024a), and more.

conditions are often tricky to identify, as they can be hidden under layers of different settings. Another commonly overlooked aspect of lossy compression is that *task accuracy is only a proxy for model utility*. Equally important is *behavioral robustness* — whether the model responds consistently — which compression often undermines. Dutta et al. (2024) show that even at relatively high bitwidths (W8A16, W8A8), models may retain similar accuracy yet undergo substantial behavioral drift. For instance, after compressing Llama2-13B with SmoothQuant to W8A8 (Xiao et al., 2023), 18.99% of MMLU answers (Hendrycks et al., 2021) flipped from correct to incorrect or vice versa, suggesting that lossy-compressed models may behave markedly differently despite comparable accuracy metrics.

In other words, **while KV cache compression offers substantial efficiency gains, deploying models that rely directly on lossy compressed KV caches often risks reduced reliability.** Given the infinitely diversified task scenarios, service owners cannot feasibly surface all such failures without constant and exhaustive stress testing. This motivates approaches that preserve lossless generation quality while still exploiting lossy compression internally. FAFO fits into this class of methods, providing such reliability while delivering meaningful latency benefits.

### 1.2. KV Cache Compression + N-Gram Candidate Pool Decoding Brings Unique Advantages over Typical Self-Speculative Decoding

**Self-speculative decoding (SD) makes SD draftless and easier to deploy** While lossy KV cache compression methods might have their own pitfalls if employed end-to-end, it is common knowledge that not every word within a perfectly written sentence is hard to infer, as plain languages like English often contain many easy filler words that do not require outstanding model intelligence to predict correctly. In fact, this very idea fuels the established Speculative Decoding (SD) paradigm (Xia et al., 2022; Leviathan et al., 2023), where a small draft model with fewer resource demands first generates guesses, which are then verified with the larger target model. The main drawback of standard SD methods is that they involve non-trivial effort and resources to align and host a separate draft model (Yan et al., 2025; Li et al., 2024b;c; 2025b). A subfield named Self-Speculative Decoding (self-SD) has since emerged by removing the need to host a separate draft model, where the draft tokens are instead generated by the same target model but under a much more resource-efficient mode. Specifically, methods like Sun et al. (2024); Sadhukhan et al. (2025); Zhang et al. (2023a); Xia et al. (2025) have explored the general idea of integrating lossy KV cache compression with self-speculative decoding, attempting to deliver lossless generation quality with improved latency.

**Self-SD's inherent shortcomings** However, **we find all such self-speculative decoding methods come with at least one of the two practical shortcomings.** First, they typically require maintaining separate sets of KV cache for the "draft" and "target" token generation, making them more memory-hungry than simply doing full model inference. This undercuts the very purpose of going draftless in the first place: self-SD methods typically deliver worse latency performance than standard SD (as the latter can afford to craft and train an SD-specific draft model (Li et al., 2025b)), so the main goal of going draftless is to reduce memory demands, which is critical in resource-constrained scenarios like local hosting. Second, we find many such self-SD methods lack general usability: for instance, TriForce (Sun et al., 2024) demands a tiny long-context model that shares the target model's vocabulary, and can be as slow as $0.17\times$ the full model inference under certain scenarios. SS (Zhang et al., 2023a) can require 7-to-20+ hours of task-specific optimization before it is ready for inference; MagicDec (Sadhukhan et al., 2025) only performs well under large batch sizes, and in practice its implementation limits its maximum generation length to just 96 tokens or lower (Wu et al., 2025). While we respect all prior art for their contributions, we believe it is fair to argue that such brittleness makes them less ready for real-world deployment.

**N-gram decoding: immediate benefits and integration challenges** With this in mind, we look into other lossless efficient decoding channels and find the n-gram candidate pool paradigm pioneered by Lookahead Decoding (Fu et al., 2024) to be a potential candidate. **Different from the sequential draft-then-verify design of SD, n-gram decoding generates its newly drafted n-grams in parallel with its lossless verification** (in Lookahead, both are done with full KV cache). This allows n-gram decoding to achieve a single KV cache footprint and, by design, completely sidesteps the first shortcoming mentioned above. We also find Lookahead to be much more robust to task scenarios in comparison to methods like TriForce (though it still breaks under certain workloads).

However, n-gram decoding has its own quirks. Most significantly, since drafting and verification occur within the same forward pass, it requires non-trivial system engineering to support any kind of KV cache compression method in a meaningful way. This is vastly different from standard SD, where the sequential pipeline allows trivial access to almost all KV cache compression methods, since one can simply engage the drafting forward passes with compression and the verification ones without. To the best of our knowledge, no prior work has successfully integrated n-gram decoding with lossy KV cache compression, resulting in some major bottlenecks — e.g., Lookahead is unable to host a large number of n-gram guesses, making its end-to-end latency

advantage less significant than the n-gram potential would otherwise allow (Fu et al., 2024; Xia et al., 2025).

**FAFO's advantages and contributions**   To bridge the gap, we present the FAFO framework, where we utilize the model with compressed KV cache to Fumble Around with great freedom and efficiency, collecting n-gram guessed tokens and storing them in a candidate pool, while simultaneously Find Out in the same forward pass with the full KV cache. We present FAFO as a general framework that is compatible with all typical static or dynamic token dropping-based KV cache compression methods as a means of generating guessed tokens. In summary, our main advantages and contributions are as follows:

- **Leveled-memory, lossless n-gram decoding.** FAFO operates under a leveled memory footprint (relative to full-model inference) while using a *single* KV cache and maintaining lossless quality. In contrast, lossy KV cache compression methods sacrifice quality, and (self) SD methods usually require enlarged or duplicated KV caches to stay lossless. These advantages are largely inherited from the n-gram paradigm and characterize FAFO's benefits over self-SD methods; to the best of our knowledge, only Lookahead Decoding shares this trifecta (Fu et al., 2024).
- **Customized KV cache manager that leverages FlexAttention.** We develop a custom KV cache manager that leverages FlexAttention (Dong et al., 2025) as an interface to connect the n-gram decoding paradigm and token dropping–based KV cache compression under our FAFO framework, enabling flexible exploration of future n-gram methods with a variety of lossy KV schemes — without the grind of custom kernel development.
- **General usability across scenarios.** We evaluate far more downstream tasks than most efficient decoding works and empirically show that FAFO is more robust than typical self-SD methods across models, tasks, input sequence lengths, etc., delivering consistent speedups with strong quality preservation. While Lookahead is already a decently robust method, FAFO enjoys a noticeable performance lead over Lookahead and remains performant under task scenarios where Lookahead is not (e.g., Table 3).
- **Revive a stagnant paradigm.** The FAFO framework opens up a new avenue of efficient draftless decoding leveraging KV cache compression techniques. We argue that this breakthrough is significant, as there has not been another lossless n-gram decoding method since the initial Lookahead Decoding, which debuted in late 2023. This contrast is especially striking, since within the same timeframe, we have seen a cluster of lossy KV cache–based SD methods developed despite their innate shortcomings (Zhang et al., 2023a; Elhoushi et al., 2024; Liu et al., 2024a; Xia et al., 2025; Sun et al., 2024; Sadhukhan et al.,

2025), calling for a revisitation of n-gram decoding to practically materialize its preferable properties.

## 2. Related Works

**Self-Speculative Decoding**   To the best of our knowledge, a few representative self-SD works have touched on the SD idea under a strict draftless context: Self-Speculative (SS) (Zhang et al., 2023a), TriForce (Sun et al., 2024)[2], MagicDec (Sadhukhan et al., 2025), and SWIFT (Xia et al., 2025). Specifically, SS and SWIFT leverage layer-skipping as the means to efficiently generate draft tokens, whereas TriForce and MagicDec integrate with KV cache compression methods like SnapKV (Li et al., 2024a) and LM-Infinite/StreamingLLM (Han et al., 2024; Xiao et al., 2024).

**FAFO differs from SD methods by only requiring one model and one set of KV cache.** Compared to the four self-SD methods like TriForce and SWIFT, FAFO prevails in requiring just one set of KV cache. This is because SD methods operate in a sequential draft-then-verify way, where some of the newly generated tokens will always rely on a lossy KV cache. We provide a detailed walkthrough of why this drawback is innate to SD methods in Appendix C. Further, **FAFO is empirically much more scenario-agnostic and overall more performant than such self-SD methods** (see Table 2). Finally, technically speaking, FAFO also differs from SD in its parallel verification process, which we will discuss in the next paragraph.

**N-Gram Candidate Pool Decoding**   Among efficient yet lossless decoding pipelines, n-gram decoding presents a unique paradigm. Developed from the Jacobi Decoding process (Santilli et al., 2023) and first proposed under Lookahead Decoding (Fu et al., 2024), n-gram decoding generates multiple guessed tokens and stores them in a candidate pool. It can then generate additional n-gram candidates and verify them against existing ones under the same forward pass in a truly parallel manner. This stands in contrast to the sequential process of all SD methods, where guessed tokens must first be generated by the draft model and only then verified by the target. We note that, while some prior art often introduces n-gram decoding (e.g., Lookahead Decoding) under the same realm as speculative decoding, **the n-gram and speculative decoding paradigms are completely different, given the parallel draft-and-verify vs the sequential draft-then-verify distinction.** Unique opportunities and challenges arise across each pipeline, where careful considerations must be made. For instance, this parallel design grants n-gram decoding several unique properties. Most notably, it allows n-gram decoding to take advantage of guessed n-gram tokens that are correct further down the

---

[2]TriForce is in fact not strictly draftless; see Appendix B for details.

decoding path but not immediately as the next decoded token — something SD cannot do (and therefore must invest decent effort to ensure the draft and the target are aligned (Li et al., 2025b; Yan et al., 2025)). This phenomenon is extremely common, as from a linguistic standpoint, some degree of repetition is often needed to form a cohesive paragraph. Such repetition is exemplified by their blog's first GIF (Figure 1), which we recommend readers check out to get an intuitive sense of how frequently this occurs.

FAFO's verification process follows this n-gram candidate pool design. However, different from Lookahead Decoding, **FAFO's guessed token generation employs a cache-compressed version of the target model, allowing it to generate many more n-gram candidates within the same forward pass.** We emphasize that although this update sounds simple, it involves non-trivial system optimization, as a naive attempt to integrate KV cache compression with n-gram decoding would fail: the expensive overhead of mask recomputation during each decoding step would cancel out any latency improvements. However, with our custom KV cache manager, such integration can be efficiently achieved, addressing key n-gram challenges like how to host a large number of n-gram guesses effectively (which bottlenecks Lookahead as demonstrated in Figure 1). Additional improvements over Lookahead are discussed in Section 4.

# 3. Why FAFO: A Practical Overview of FAFO's Advantages

This section provides a brief overview of why FAFO is worth considering from a practical standpoint, with the goal of justifying its design choices through small controlled comparisons; the full method, attention-mask design, and KV cache management are provided in Section 4.

As detailed in Section 2, KV cache compression and self-speculative decoding have already seen mature development, while n-gram decoding — though stagnant in terms of progress — is also an established paradigm (Fu et al., 2024). Thus, much of the contribution of our work lies in whether our proposed FAFO can present significant advantages over these existing solutions, which we shall gladly report that it can.

We organize the discussion around the two main comparative paradigms (self-SD methods and Lookahead Decoding) and show why FAFO's design addresses pain points that are inherent to these schools of work. First, self-SD methods (Section 3.1): their sequential draft-then-verify pipeline forces them to hold multiple sets of KV cache and to align draft and target predictions at exact positions, which is fragile under multi-turn or task-mismatched workloads. FAFO sidesteps both issues by keeping only one KV cache and verifying n-grams in parallel. Second, Lookahead Decoding

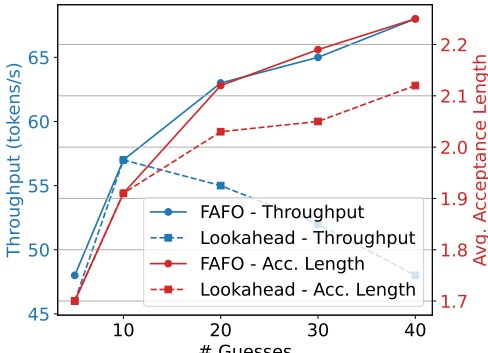

*Figure 1.* FAFO vs Lookahead Decoding with increased number of n-gram guesses. FAFO significantly outperforms in terms of both practical speedup and theoretical n-gram generation quality.

(Section 3.2): although it shares the n-gram candidate-pool design, its practical speedup degrades once the number of guesses grows — as already noted by its own authors. FAFO removes that ceiling by generating guesses on a compressed cache and caching them by longer suffixes. We elaborate on each of these in turn below.

## 3.1. FAFO Presents Significant Memory Savings over Self-Speculative Decoding Methods

Self-SD methods reduce memory footprint compared to standard SD by eliminating the need for a separate draft model. Naturally, self-SD methods like TriForce (Sun et al., 2024) and SWIFT (Xia et al., 2025) tend to leverage compressed KV cache for draft token generation. However, they still require hosting multiple sets of KV cache, as the draft forward must be computed independently from the target forward with full cache (see Appendix C for details).

*Table 1.* Llama-2-7b-chat over MT-Bench, with TriForce additionally utilizing a llama-68m as the draft-draft model, following its own configuration. $\tau$ is the average acceptance length, a theoretical upper bound of practical speedup if no overhead is considered.

| Method | Peak Memory (MB) | Speedup | $\tau$ |
|---|---|---|---|
| FAFO-Stream | **2362** | **1.91×** | 2.29 |
| Lookahead | **2183** | **1.61×** | 1.66 |
| FAFO-2forward | 3455 | 1.19× | 2.18 |
| TriForce | 4584 | 0.21× | 1.06 |
| SWIFT | 6196 | 1.17× | 2.65 |

Table 1 shows that TriForce occupies a much larger memory footprint than FAFO. We also observe that under challenging multi-turn tasks like MT-Bench (Zheng et al., 2023), TriForce is significantly slower than simply running inference on the full model. This is because it requires guessed tokens to be generated at exactly the right time and position — a condition that is difficult to meet, as reflected by the $\tau = 1.06$ theoretical upper bound in Table 2. In contrast, FAFO with StreamingLLM (Xiao et al., 2024) as the

backbone KV cache compression method achieves a $1.91\times$ practical speedup with a $2.29\times$ theoretical upper bound — a significant improvement over TriForce.

To determine whether FAFO's gains over TriForce stem from the n-gram candidate pool allowing more flexible guessed token positions than speculative decoding, or from KV cache compression being more effective under parallel verification, we design an investigatory method called FAFO-2forward (Table 1). In FAFO-2forward, we drop the parallel verification design; instead, we perform one forward pass with compressed KV cache to collect n-gram guessed tokens, followed by a second forward pass to verify. Results show that FAFO-2forward is outperformed by the full FAFO-Stream by 60% ($1.91\times$ vs. $1.19\times$) in practical latency speedup, despite having similar theoretical average acceptance lengths ($\tau = 2.29$ vs. $2.18$). This indicates that while maintaining an n-gram candidate pool does provide an inherent advantage over strict sequential speculative generation when it comes to integration with KV cache compression means, the pipeline is only fully leveraged when combined with parallel verification upon fumble-generated tokens, justifying the integrated framework by FAFO.

### 3.2. FAFO Allows for Much More Guessed N-Gram Generation than Lookahead Decoding

Recall that Lookahead is the only prior n-gram candidate-pool method; we briefly review its mechanics before contrasting FAFO. At a high level, Lookahead runs a single forward pass that simultaneously refines a window of speculative future tokens (the *lookahead branch*), harvests $n$-grams from that window into a candidate pool, and verifies previously harvested $n$-grams against the model's true distribution (the *verification branch*) — making drafting and verification parallel rather than sequential (readers unfamiliar with these mechanics are referred to Appendix B.1 for a self-contained walkthrough).

One major distinction between FAFO and Lookahead Decoding is that FAFO's n-gram guessed tokens are generated using only compressed KV cache (Section 4.2). This design allows FAFO to generate far more guesses than Lookahead, overcoming a key limitation recognized by the Lookahead authors (see Figure 8 of (Fu et al., 2024), where Lookahead's practical speedup drops sharply as the number of guesses increases). As shown by the blue lines in Figure 1, FAFO can generate many more n-gram guesses without incurring a speed penalty, whereas Lookahead peaks around 10 guesses and then suffers decreasing throughput as the number of guesses increases. Note that the number of hosted n-gram guesses almost directly determines the overall performance of a candidate pool-based method, since more guesses provide more opportunities for matches.

Beyond Fumble Decoding, we also introduce a small but useful improvement at the verification stage: Find Out Verification (Section 4.3). Instead of caching n-grams solely based on the most recently generated token, we assign a lookback window and cache them as part of the n-gram. Combined with Fumble Decoding, this enables FAFO to generate both more numerous and higher-quality n-grams, as reflected by the red lines in Figure 1, which show the theoretical upper bound of the average token acceptance length.

## 4. FAFO: Fumble Around and Find Out

FAFO implements a **unified execution pipeline** that decouples token generation logic from physical memory constraints. Specifically, the system orchestrates two concurrent workloads: (1) *Fumble Around*, which utilizes a compressed KV cache to generate high-throughput speculative guesses in parallel; and (2) *Find Out*, which verifies these previously generated guesses against the full set of KV cache. To maximize GPU utilization, FAFO fuses these operations into a *single forward pass* via a custom sparse attention kernel (Section 4.4). This design amortizes the cost of loading QKV blocks over both drafting and verification, eliminating the synchronization overhead of separate draft/verify kernels typical in standard speculative decoding, while enabling multi-token acceptance per step (Figure 2).

### 4.1. Preliminary

We consider a language model $p$, and a full sequence of tokens available at a given point in decoding $x_{1:|x|}$, consisting of both the initial prefill tokens and all tokens generated so far, where $|x|$ denotes the total number of tokens. Associated with $x_{1:|x|}$ is a set of key-value (KV) cache entries, denoted by $\text{kv}_{1:|x|}$. Let $y^i_{s_i+1:s_i+k} = (y_{s_i+1}, y_{s_i+2}, \ldots, y_{s_i+k})$ denote a subsequence of $k$ tokens, where the index range $(s_i+1:s_i+k)$ refers to *absolute positional indices*, assuming the tokens in subsequence $i$ are placed at positions $s_i$ later in the current decoded sequence $x_{1:|x|}$. FAFO maintains a set of $n$ such subsequences as speculative future subsequences. We also define a KV cache compression function $\mathcal{C}$, i.e., methods such as StreamingLLM (Xiao et al., 2024).

### 4.2. Fumble Decoding

FAFO uses a compressed set of KV cache to speculate future tokens. At each decoding step, FAFO maintains a set of $n$ independent speculative subsequences $y^1_{s_1+1:s_1+k}, y^2_{s_2+1:s_2+k}, \ldots, y^n_{s_n+1:s_n+k}$. Given the current KV cache entries $\text{kv}_{1:|x|}$ and a KV cache compression function $\mathcal{C}$, FAFO leverages the compressed cache $\mathcal{C}(\text{kv}_{1:|x|})$ to generate the next token for each speculative subsequence $y^i$ in parallel with negligible memory bandwidth cost as:

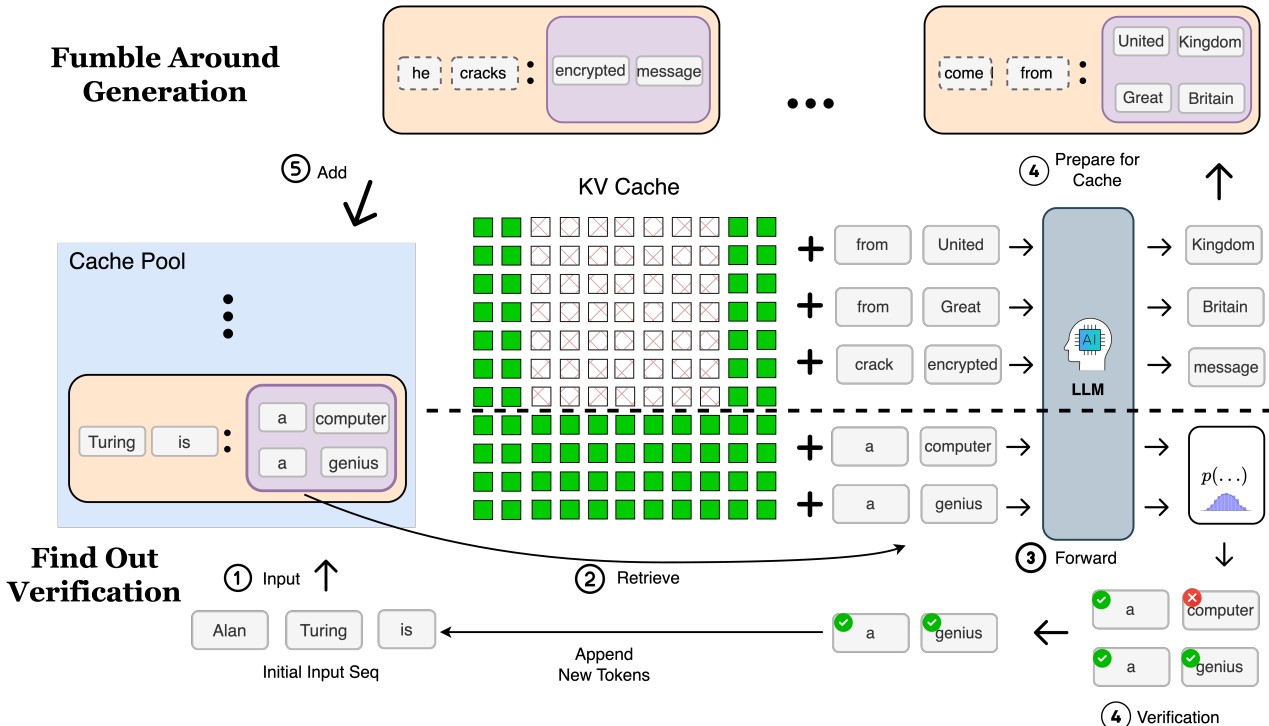

*Figure 2.* General Pipeline of FAFO. **Walkthrough of FAFO decoding** with (1) the input prompt *"Alan Turing is"*. (2) FAFO retrieves 2-gram candidate sequences such as *"a computer"* and *"a genius"* based on the suffix *"Turing is"*. (3) In parallel, Fumble Decoding generates next tokens of guesses like *"from United"*. A single forward pass is executed for both the Fumble Decoding and Find Out Verification branches. (4.1) Verified candidates (e.g., *"a genius"*) are accepted and appended to the input. (4.2) Newly generated 2-grams (e.g., *"United Kingdom"*) are cached by combining with previously buffered tokens from the same speculative subsequence (e.g., *"come from"*). (5) These new cached sequences are then added to the shared cache pool for reuse.

$$y^i_{s_i+k+1} = \text{argmax}\, p(y^i_{s_i+k+1}|y^i_{s_i+1:s_i+k}, \mathcal{C}\left(\text{kv}_{1:|x|}\right))$$

To ensure continuous generation of future tokens, the oldest token in each subsequence, namely $y^i_{s_i+1}$, is discarded after every decoding step. The remaining subsequence $y^i_{s_i+2:s_i+k+1}$ is then collected and offloaded to a CPU-side pool for later use in the "Find Out" verification phase with FAFO's prefix caching strategy.

**FAFO's prefix caching strategy.** For each guess stream $i$, FAFO maintains a buffer of the last $k$ discarded tokens, $y^i_{s_i-k+2:s_i+1}$, and uses this buffer to organize how newly generated subsequences are cached. Each new guess subsequence $y^i_{s_i+2:s_i+k+1}$ is cached under multiple discarded-token prefixes of increasing length up to $k$ (Algorithm 2):

$$(y^i_{s_i+1}), \quad (y^i_{s_i}, y^i_{s_i+1}), \quad \ldots, \quad (y^i_{s_i-k+2:s_i+1}).$$

Later, during verification, when the current decoding context ends with a particular prefix, the system retrieves all cached subsequences associated with that prefix and reuses them as candidate guesses. Intuitively, longer matching prefixes share more tokens with the current context, so the corresponding candidates have a higher probability of matching

the model's distribution. Proposition E.1 provides intuition for why this strategy is effective: longer matching prefixes correlate with higher suffix probability, suggesting that candidates indexed under longer prefixes are more likely to be accepted. Thus, the verification phase prioritizes candidates indexed under longer prefixes, improving the quality of verified guesses while keeping the guess pool entirely on the CPU and avoiding additional GPU memory overhead.

### 4.3. Find Out Verification

FAFO retrieves $n$ guesses (speculative subsequences) $a^1, \ldots, a^n$, previously generated by Fumble Around decoding, from the CPU-side cache pool based on the current decoded prefix (Alg. 3; Fig. 2). Concretely, FAFO considers up to $k$ of the most recent tokens in the current decoded sequence, $x_{|x|-m+1:|x|}$ for $1 \le m \le k$, and uses these length-$m$ suffixes as lookup keys into the cache, starting from the longest $m$ for which entries exist until we have retrieved enough $n$ guesses. Each cached guess was originally indexed under the corresponding discarded-token prefix when it was generated, so this lookup returns candidates whose prefix context matches the current decoding state as closely as possible.

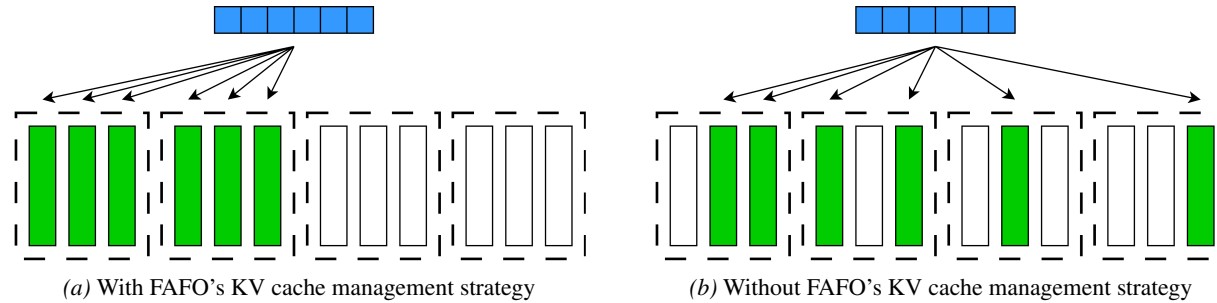

*(a)* With FAFO's KV cache management strategy          *(b)* Without FAFO's KV cache management strategy

*Figure 3.* **Logical Sparsity vs. Physical Fragmentation.** (a) **With FAFO:** By actively compacting relevant tokens (green) into a dense region, FAFO ensures that logical sparsity translates to reduced physical I/O. The kernel iterates only over the active block budget $\mathbf{B}_{fixed}$, effectively terminating the memory load early. (b) **Without FAFO:** Relevant tokens are fragmented across the address space. Because FlexAttention loads data at **block granularity**, the hardware performs redundant HBM transactions on blocks containing mostly irrelevant (white) data, limiting bandwidth utilization.

The retrieved candidates are then verified in parallel using the full KV cache via distribution matching, following the standard speculative- decoding accept/reject rule of (Leviathan et al., 2023): the target LLM is forwarded on the draft tokens, and each token is accepted if and only if the model's predicted next token exactly matches the draft token (Alg. 4). Unlike Lookahead, which caches and retrieves candidates conditioned solely on $x_{|x|}$, FAFO conditions retrieval on a longer suffix, yielding higher-quality guesses. After verification, all accepted tokens are appended at once before the next iteration, and in practice more than one token is typically accepted, translating into lower end-to-end decoding latency.

### 4.4. System-Efficient FAFO via Sparse Attention Kernels

FAFO implements a *draftless* pipeline by fusing the Fumble Around (speculation) and Find Out (verification) phases into a single model forward pass. By concatenating draft tokens and verification tokens, we leverage a unified attention mask (Figure 4) to maximize GPU utilization. However, simply masking tokens at the logical level does not guarantee system efficiency, as standard attention kernels still load the full KV cache from memory. To overcome this, we employ FlexAttention to enforce block-sparse computation, but its naive application introduces new overheads.

**The Challenge of Dynamic Masking.** While FlexAttention is well-suited for the prefill phase, where the input length remains static, applying it to the decoding phase is highly non-trivial. The context grows with each generated token, requiring the block mask to be recomputed at every step. **This recomputation is prohibitively costly, rendering naive use of FlexAttention during decoding impractical** without a decoding-aware block masking scheme specifically designed to handle dynamic attention contexts. Specifically, dynamic mask regeneration triggers *repeated graph capture and compilation overheads*, effectively negating the computational benefits of sparsity by stalling the GPU pipeline.

**Hardware-Aware KV Layout for Zero-Overhead Masking.** We resolve this by decoupling the *logical* position of tokens from their *physical* memory location. We implement a **Physically Contiguous, Logically Sparse** memory management strategy. We allocate a fixed buffer of physical KV blocks, $\mathbf{B}_{fixed}$, at the head of the KV tensor to store the compressed entries $\mathcal{C}(\text{kv}_{1:|x|})$.

$$\text{KV Cache Physical Layout} = \left[ \underbrace{\mathcal{C}(\text{kv}_{1:|x|})}_{\text{Selected Blocks}} \parallel \underbrace{\text{R}(\text{kv}_{1:|x|})}_{\text{Irrelevant Blocks}} \right]$$
(1)

By enforcing that all relevant tokens for the current "Fumble Around" step reside within $\mathbf{B}_{fixed}$, we construct a static FlexAttention block mask that instructs the kernel to load *only* the first $|\mathbf{B}_{fixed}|$ blocks from HBM to SRAM. Irrelevant entries $\text{R}(\text{kv})$ are logically preserved but physically skipped during the attention kernel execution.

**Inference KV Cache Management.** Between decoding steps, we execute an asynchronous copy to swap newly selected tokens by function $\mathcal{C}$ into $\mathbf{B}_{fixed}$ and evict expired tokens to the tail (Appendix G). This guarantees that the FlexAttention kernel operates on a dense, contiguous memory region. In contrast, the vanilla implementation suffers from severe block-level fragmentation, where valid KV entries are sparsely distributed across the physical address space. Because FlexAttention loads data at the granularity of blocks, this fragmentation forces the kernel to incur redundant HBM transactions, fetching entire physical blocks even when they contain negligible valid context (Figure 3). By enforcing a dense memory layout that eliminates the overhead of dynamic mask regeneration, FAFO maximizes effective memory bandwidth utilization, directly translating logical sparsity into realized wall-clock acceleration.

## 5. Experiments

**Models and Settings** Our core evaluations focus on Llama-2-Chat (7B) (Touvron et al., 2023), LLaMA-3-

*Table 2.* Observed practical latency speedup and average acceptance length $\tau$ on MT-bench, GSM8K, and HumanEval-Completion.

| Models | Method | MT-bench | | GSM8K | | HumanEval-C | |
|---|---|---|---|---|---|---|---|
| | | Speedup | $\tau$ | Speedup | $\tau$ | Speedup | $\tau$ |
| **Llama-2-7b-chat** | FAFO-Stream | **1.91**× | 2.29 | **1.63**× | 2.70 | **2.03**× | 2.34 |
| | FAFO-Quest | 1.32× | 2.20 | 1.40× | 2.60 | 1.63× | 2.33 |
| | Lookahead | 1.61× | 1.66 | 1.58× | 1.65 | 1.72× | 1.77 |
| | TriForce | 0.21× | 1.06 | 0.17× | 1.12 | 0.22× | 1.06 |
| | SWIFT | 1.17× | 2.65 | 1.22× | 2.43 | 1.13× | 3.79 |
| **Llama-3-8B-Instruct** | FAFO-Stream | **1.58**× | 2.12 | **1.60**× | 2.10 | **1.65**× | 2.00 |
| | FAFO-Quest | 1.50× | 2.05 | 1.45× | 2.05 | 1.43× | 2.00 |
| | Lookahead | 1.49× | 2.01 | 1.39× | 1.99 | 1.52× | 1.90 |
| | SWIFT | 1.18× | 3.33 | 1.34× | 3.73 | 1.22× | 3.90 |
| **Llama-3.1-8B-Instruct** | FAFO-Stream | **1.50**× | 2.10 | **1.31**× | 2.08 | **1.57**× | 2.20 |
| | FAFO-Quest | 1.40× | 2.08 | 1.28× | 2.08 | 1.48× | 2.19 |
| | Lookahead | 1.21× | 2.01 | 1.25× | 2.01 | 1.28× | 2.10 |
| | SWIFT | 0.94× | 2.93 | 0.98× | 3.22 | 1.03× | 2.38 |
| **Qwen2.5-7B-Instruct** | FAFO-Stream | 1.43× | 2.10 | **1.60**× | 2.30 | **1.44**× | 2.10 |
| | FAFO-Quest | **1.46**× | 2.20 | 1.44× | 2.20 | 1.38× | 2.08 |
| **Qwen2.5-32B-Instruct** | FAFO-Stream | **1.20**× | 2.07 | 1.30× | 2.36 | **1.40**× | 2.33 |
| | Lookahead | 1.09× | 2.15 | **1.44**× | 2.42 | 1.26× | 2.36 |

Instruct (8B) (Grattafiori et al., 2024), LLaMA-3.1-Instruct (8B) (Grattafiori et al., 2024), and Qwen2.5-Instruct (7B, 32B) (Yang et al., 2024), providing fair coverage of different attention architectures, model families, and scales. Additionally, we evaluate DeepSeek-R1-Distill (Qwen-7B and Llama-8B) (Guo et al., 2025) under reasoning tasks. All experiments are conducted on a single NVIDIA A100 GPU with 80GB of memory. Unless otherwise specified, all models are served with FP16 precision and a batch size of 1, following the setup of existing latency-oriented works (Cai et al., 2024; Fu et al., 2024). FlexAttention kernels (Dong et al., 2025) with our customized KV cache manager are used for efficient sparse attention computation.

**Benchmarks and Metrics** Following prior speculative decoding works (Fu et al., 2024; Li et al., 2024b), we evaluate FAFO on three widely used benchmarks: MT-Bench (Zheng et al., 2023), GSM8K (Cobbe et al., 2021), and HumanEval (Chen et al., 2021). Additionally, we test FAFO on Multi-IF (He et al., 2024), SCBench (Li et al., 2025a), LongBench (Bai et al., 2023), and AIME 24 to further demonstrate performance under multi-turn, long-context, and long-generation scenarios. Finally, we include PG19 for alignment with TriForce. Following Fu et al. (2024); Li et al. (2024b); Sadhukhan et al. (2025), we report FAFO's performance with the following metrics:

- **Wall-clock speedup ratio**: The observed speedup relative

to vanilla autoregressive decoding, measured in tokens per second.

- **Average acceptance length** $\tau$: The average number of tokens accepted per decoding step.

**Baselines** We compare FAFO against established self-SD and n-gram decoding methods, including TriForce (Sun et al., 2024), SWIFT (Xia et al., 2025), and Lookahead Decoding (Fu et al., 2024). We omit MagicDec (Sadhukhan et al., 2025), as its implementation cannot support more than 96 newly decoded tokens, making it incompatible with most challenging tasks. Older methods like SS (Zhang et al., 2023a) are also omitted, as they have been extensively compared with our featured baselines and no longer represent the state of the art (Xia et al., 2025). "FAFO-Stream" and "FAFO-Quest" refer to our proposed FAFO method instantiated with LM-Infinite/StreamingLLM (Han et al., 2024; Xiao et al., 2024) and Quest (Tang et al., 2024) as the KV cache eviction methods. We chose these two because they are established representatives of static and dynamic token-dropping approaches. **While we could have developed a custom KV cache compression method to replace them, we intentionally avoid doing so** to respect prior art and to prevent reinventing the wheel under unnecessary conditions. More importantly, we believe the community benefits most from a general framework where they can experiment with different KV cache compression methods, rather than

*Table 3.* Speedup ratio and average acceptance length $\tau$ on datasets from LongBench.

| Models | Method | Multi-News | | LCC | | TREC | | Qasper | | 2WikiMQA | |
|---|---|---|---|---|---|---|---|---|---|---|---|
| | | Speedup | $\tau$ | Speedup | $\tau$ | Speedup | $\tau$ | Speedup | $\tau$ | Speedup | $\tau$ |
| **L3.1 8B** | FAFO-Stream | **1.97**× | 2.81 | **1.78**× | 3.08 | **2.01**× | 4.27 | **2.20**× | 3.54 | **1.94**× | 3.73 |
| | FAFO-Quest | 1.62× | 2.83 | 1.70× | 3.10 | 1.80× | 4.37 | 2.12× | 3.65 | 1.87× | 3.95 |
| | Lookahead | 1.12× | 1.93 | 1.48× | 2.58 | 1.54× | 3.69 | 0.85× | 2.19 | 1.01× | 2.94 |
| | SWIFT | 1.09× | 3.15 | 1.08× | 3.72 | 1.09× | 4.17 | 1.41× | 4.10 | 1.13× | 3.25 |

a baked implementation supporting only our own.

**End-to-End Effectiveness.** Table 2 reports the wall-clock speedup ratio and average acceptance length of FAFO compared to other baselines. FAFO-Stream or FAFO-Quest essentially achieves the highest practical speedup and $\tau$ under all featured settings, approximately 30% faster than Lookahead Decoding. Specifically, we find FAFO to be robust under challenging tasks like MT-Bench and GSM8K — tasks that pose a significant challenge to methods like TriForce and SWIFT, which might experience negative speedups.

Given that KV cache is most significant under a long context setting, we further feature some common tasks from LongBench to confirm FAFO's task robustness. According to Table 3, we are glad to report that FAFO tends to offer even better performance on long context tasks. This result is intuitive, as constant-budget KV cache compression methods like LM-Infinite/StreamingLLM often offer the most efficiency gains under such long context settings.

## 6. Conclusion and Limitations

While FAFO surely brings significant practical speedup and revives the stagnant n-gram decoding paradigm, it does come with several limitations. One limitation of FAFO, and broadly n-gram decoding methods, is the lack of effective support for batched inference (which is also recognized by (Fu et al., 2024)); thus, we experiment with FAFO in latency-sensitive scenarios (batch size = 1). While draftless methods are naturally well-suited to local deployment and resource-constrained scenarios where batching is not a primary concern, batch capability would nonetheless be advantageous. We benchmark the per-step FLOPs of FAFO and Lookahead Decoding (Fu et al., 2024) and observe that n-gram methods incur substantially higher computational costs than vanilla decoding. This is intuitive, as unlike speculative decoding, which only requires target verification once in a while, n-gram methods both verify and generate multiple guesses within every forward pass, leading to significantly higher FLOPs. As a result, n-gram decoding is more likely to become compute-bound in batched settings, limiting potential speedups. We do not claim that batched n-gram decoding is infeasible; rather, enabling efficient batching would require substantial redesign and is an important direction for future

work. We highlight this limitation to provide transparency and encourage practitioners to consider whether their target deployment scenario aligns with the n-gram paradigm.

Another common criticism of FAFO is its algorithmic novelty (or lack thereof). We discuss this in Appendix A.

## Impact Statement

Since our method is theoretically lossless — preserving the original model's output distribution by construction — we do not believe it raises direct concerns regarding ethical or societal consequences stemming from altered model behavior. We note that this distinction is worth highlighting for lossy acceleration methods, where compressed representations can silently shift model outputs in ways that may affect downstream safety and fairness properties.

That said, even lossless methods can in practice yield minimally different results compared to their uncompressed counterparts, primarily due to non-determinism in floating-point execution order. We therefore advise practitioners to thoroughly evaluate our method under their specific tasks, particularly if it is to be adopted in critical or high-stakes scenarios.

Additionally, n-gram candidate-pool decoding paradigms such as FAFO incur higher per-step computational cost in exchange for improved throughput. Depending on the provider's infrastructure and serving strategy, this trade-off can result in increased energy consumption and greenhouse gas emissions relative to standard autoregressive decoding.

## Acknowledgements

This research is kindly supported by Visa Research. We thank Yuzhong Chen for his supportive mentorship and strong technical guidance throughout this project. We also thank the anonymous reviewers for their constructive feedback.

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

# A. Extended Limitations

**Batch Support** One major limitation of FAFO — or n-gram decoding methods in general — is the lack of batch inference support; all experiments are done under the most extreme latency-sensitive setup (batchsize = 1). While we do believe that draftless methods, by nature, make the most sense in local deployment or resource-constrained scenarios where batching is not a top priority, we also recognize that having such support would be a clear advantage.

We benchmarked the per-step FLOPs of both FAFO and Lookahead Decoding (Fu et al., 2024) and observed that n-gram methods typically incur substantially higher FLOPs than vanilla decoding. In contrast, speculative decoding (SD) methods, given their sequential draft-then-verify design, often exhibit lower average FLOPs than the original model (except under extreme draft model configurations). The high FLOPs consumption of n-gram methods is intuitive: instead of relying on a smaller draft model (as in SD) or performing a single full model forward pass (as in vanilla decoding), n-gram decoding performs full model verification *and* generates multiple new n-gram candidates within the same forward pass. This implies that in a batched setting, n-gram methods are likely to hit the compute-bound regime much sooner, leaving less room for meaningful speedup.

For disambiguation, we are not claiming that batched n-gram decoding is impossible. In fact, we believe there are many promising avenues for improving the compute and memory efficiency of the n-gram pipeline established by Lookahead (e.g., one of them is to avoid concatenating n-gram guesses and use an "inner batch" mechanism with shared prefix cache to verify multiple guesses). Our point is simply that **developing a truly batch-capable n-gram method would require substantial new systems design, likely worthy of a dedicated follow-up paper.** We provide this section here to be transparent with our readers about this current limitation shared by FAFO, Lookahead, and the general n-gram paradigm. The readers are advised to carefully gauge whether the intended task scenario fits the n-gram paradigm's advantages before adopting our method.

**Algorithmic Novelty** We acknowledge that FAFO is more integration- and engineering-driven than algorithmically novel. However, we argue that it remains highly contributive within its specific research landscape, as algorithmic novelty is only one aspect of a method.

- **Limited algorithmic freedom under hard constraints.** FAFO adopts lossy KV-cache compression under an $n$-gram framework and does not introduce many major new algorithmic primitives. That said, algorithmic novelty is only one axis of contribution. Our design space is extremely restrictive: *training-free + draftless + lossless decoding*. While highly practical, these constraints rule out common avenues for novelty, including custom training, architectural changes, draft–target co-design, or lossy accuracy–efficiency trade-offs.

- **Reuse of mature components is standard in this space.** Prior art operating under similar constraints — such as TriForce (Sun et al., 2024), MagicDec (Sadhukhan et al., 2025), and SWIFT (Xia et al., 2025) — also integrates established KV-cache compression techniques with speculative or candidate-based decoding. Careful reuse and integration of known components is therefore the norm rather than an anomaly under such user-friendly constraints.

- **The field is not saturated with arbitrary repackaging.** There has been *no* lossless and draftless successor to Lookahead since its debut in late 2023, despite a surge of lossy KV-cache and speculative-decoding methods over the past two years. If many lossless KV + $n$-gram methods already existed, FAFO could indeed be dismissed as arbitrary recombination. The absence of such follow-up instead suggests that progress in this space is nontrivial.

- **FAFO revives a stagnated research direction.** We believe it is fair to claim that FAFO practically revives the $n$-gram candidate-decoding paradigm. This indicates that meaningful progress here requires deep understanding, careful component selection, and meticulous execution, rather than random swapping of modules.

- **Substantial and exceptional performance gains.** FAFO achieves exceptional speedups. It resolves Lookahead's key limitation — its inability to host a large number of $n$-gram guesses — and delivers unmatched end-to-end acceleration under strict constraints across a broad suite of tasks. Within the efficiency community, simplicity paired with strong empirical impact is widely valued.

- **Integration-driven contributions are well precedented.** Rerouting well-understood components to solve a strictly harder problem is a legitimate and impactful contribution. Many influential efficiency works are algorithmically light but integration-savvy:
  - **QLoRA** (Dettmers et al., 2023): 4-bit quantization + LoRA, now foundational for memory-constrained PEFT.

- **KIVI** (Liu et al., 2024b): per-channel quantization + sliding-window buffering, foundational for KV-cache quantization.
- **MEND** (Mitchell et al., 2021): gradient decomposition + editing, foundational for large-scale LLM editing.

While we do not claim FAFO to be as impactful as these established works, we believe its contributions fall in a similar categorical avenue.

Last, while we might be able to score some algorithmic novelty points with a custom attention design, we intentionally prioritize extensibility over bespoke mechanisms. We believe the community benefits more from a flexible `FlexAttention`-style interface that enables many lossy KV-cache methods than from hardcoding a single unique approach purely to score novelty points.

We are firm believers in this particular guideline:

> *"The goal is to solve the problem, not to solve it in a complex way. Simpler solutions are preferable—they are less brittle and easier to deploy."*

And we hope our audience will agree.

## B. Lookahead Decoding and Extended Related Works

### B.1. A Walkthrough of Lookahead Decoding

Since FAFO inherits its overall scaffolding from Lookahead Decoding (Fu et al., 2024), we provide here a self-contained walkthrough of how Lookahead operates. Lookahead builds on the observation, first articulated by Jacobi Decoding (Santilli et al., 2023), that the next several decoding steps of an autoregressive LLM can be *posed as a fixed-point problem and refined in parallel* rather than executed one token at a time. Concretely, at every decoding step, Lookahead runs a single forward pass that simultaneously: (i) refines a window of speculative future tokens (the *lookahead branch*), (ii) harvests $n$-grams from that window into a candidate pool, and (iii) verifies stored $n$-grams against the model's true distribution (the *verification branch*).

**Lookahead branch.** Lookahead maintains a $W \times N$ "lookahead window" of token positions ahead of the current decoded prefix. Each row of the window holds a tentative continuation of length $N-1$, initialized from random or recent tokens. In one forward pass, every position in this window is updated in parallel based on the previous decoding step's logits, much like a Jacobi iteration on the autoregressive map. After the update, each row provides $N-1$ overlapping $n$-grams that are added to the candidate pool, keyed by their first token. None of these positions are accepted as output — they exist solely to harvest candidate $n$-grams cheaply, alongside the verification work that the model needs to do anyway.

**Verification branch.** In the same forward pass, Lookahead also verifies previously harvested $n$-grams. For each candidate $n$-gram whose first token matches the most recently decoded token $x_{|x|}$, Lookahead lays out the candidate's remaining tokens in additional sequence positions and uses a custom attention mask so that each candidate position attends only to the prefix plus its own ancestors within the candidate (and not to the lookahead window or to other candidates). The model's outputs at these positions are then compared against the candidate tokens, and the longest prefix that matches greedy decoding is accepted, advancing the decoded prefix by multiple tokens. Crucially, the lookahead branch and the verification branch share the same forward pass — making drafting and verification *parallel* rather than sequential.

**N-gram pool.** The candidate pool is keyed by a single token: an $n$-gram is looked up whenever its first token equals the current $x_{|x|}$. Because verification reuses an attention pass the model would have run anyway, the marginal cost of checking many candidates is small, but practical throughput degrades once the pool grows beyond what the verification mask can fit per step. This is the limitation the Lookahead authors discuss in their Figure 8, and is one of the bottlenecks FAFO targets.

**Connection to FAFO.** FAFO retains the parallel draft-and-verify-in-one-forward-pass structure but modifies two pieces. *Fumble Decoding* generates the lookahead window's tokens using a *compressed* KV cache rather than the full cache, freeing budget to host substantially more $n$-gram candidates per step (Section 4.2). *Find Out Verification* keys the candidate pool on

a longer suffix than just $x_{|x|}$, yielding higher-quality matches (Section 4.3). Readers who prefer a visual walkthrough may consult the animated Figure 4 of the Lookahead Decoding blog post.[3]

## B.2. Extended Related Works

**Token Dropping-based KV Cache Compression**   Due to the growing nature of the KV cache, many lossy compression techniques have been developed to reduce memory footprint and improve generation latency. For instance, LM-Infinite/StreamingLLM (Han et al., 2024; Xiao et al., 2024) preserves only the first few "attention sink" and recent tokens while dropping intermediate ones, achieving a constant KV cache budget. H2O (Zhang et al., 2023b) and SnapKV (Li et al., 2024a) evict tokens based on attention scores, representing static KV cache eviction methods. Dynamic counterparts like Quest (Tang et al., 2024) and NSA (Yuan et al., 2025) select retained tokens at each decoding step, unlike static methods which typically evict in one shot after prefill. More recent dynamic sparse-attention methods such as Sketch&Walk (Le et al., 2026) and SOCKET (Joshi et al., 2026) share the same motivation, selecting which tokens to attend to via lightweight per-step approximations (sketching/walk-based block scoring and LSH-based soft collision scoring, respectively) rather than persistent eviction. **FAFO differs from these methods by offering lossless generation quality.** To our knowledge, nearly all KV cache compression techniques are lossy, with their failure patterns occasionally revealed via benchmarks like `longctx_bench` (Yuan et al., 2024) and SCBench (Li et al., 2025a). FAFO is well-suited for cases demanding both latency speedup and lossless outputs.

**Speculative Decoding**   Speculative Decoding (SD) uses a smaller draft model to generate guessed tokens and a larger target model to verify them, enabling the potential of confirming multiple tokens per single forward pass (Xia et al., 2022; Leviathan et al., 2023). Later works like SpecInfer (Miao et al., 2023) and Sequoia (Chen et al., 2024) use tree attention for efficient multi-token verification.

As mentioned in Section 1, the main criticism of SD is the need to craft and host a separate draft model, which induces significant alignment efforts and resource demands — a major challenge for `r/LocalLLaMA`-like local hosting users. Thus, scholars have explored the potential of draftless SD, often known as *Self-Speculative Decoding* (self-SD), which adopts the same model as both draft and target, often with the draft forward being a sparse variant of the target forward. A concurrent and closely related work is SparseSpec (Zhao et al., 2025), which similarly pairs self-speculation with KV cache compression, but follows the standard sequential *draft-then-verify* pipeline (`spec_stride`=8 autoregressive draft steps with a compressed cache, then one verification forward pass) — the same SSD structure that FAFO's parallel n-gram candidate-pool design departs from. SparseSpec further proposes *Pillar attention*, a compression kernel that reuses verification-time attention scores to pick top-$K$ salient tokens. FAFO is instead agnostic to the underlying compression method (e.g., StreamingLLM (Xiao et al., 2024), Quest (Tang et al., 2024), or a Pillar-style scheme could all serve as its "fumble" backend) and primarily targets the batch-size-1 latency-sensitive regime rather than batched reasoning-model rollouts.

**Strictly Draftless Methods**   By "strict draftless context" in Section 2, we mean that there is one original model, with no parameter or architectural modification, serving as both draft and target. We clarify that TriForce (Sun et al., 2024) does not fit this description, as it is a three-level method where a tiny draft-draft model sharing the target model's vocabulary is used to conduct the first drafting, which is then processed by the target model with partial cache, and finally verified with the target model in full cache. Strictly speaking, TriForce is therefore not fully draftless. However, we include it here and use it as a major baseline because: a) if we remove the tiny draft-draft model, it is strictly draftless in the second and third stages, meaning that this additional stage is likely of significant value despite its small additional footprint; and b) TriForce was developed by the same lab as MagicDec (Sadhukhan et al., 2025), but is more performant under latency-sensitive (batchsize=1) scenarios, making it a major landmark to benchmark against FAFO. Hereinafter, we might refer to TriForce as a draftless/self-SD method for concise delivery.

**More Tangentially-Related Speculative Decoding Methods.**   We note that there are additional SD methods that also leverage lossy compression, such as Kangaroo (Liu et al., 2024a) and LayerSkip (Elhoushi et al., 2024). However, Kangaroo requires training additional components on top of the original model, and LayerSkip demands weight updates. As a result, they either diverge from the "strictest draftless context," or are no longer lossless to the original model. These works are technically unrelated to FAFO by and large, but we opt to feature them here because the distinction relies on an intricate

---

[3]Figure 4 of https://lmsys.org/blog/2023-11-21-lookahead-decoding/.

understanding of such methods. We also distinguish FAFO from ANPD (Ou et al., 2024), another draftless lossless n-gram decoder: ANPD draws draft tokens from a token-level statistical (Markov) n-gram model — updated by frequency counts over the prompt and already-generated tokens — and uses the standard sequential draft-then-verify pipeline. FAFO instead generates guesses by running the target model itself under a compressed KV cache and verifies them in parallel within the same forward pass, so its guesses reflect the model's own forward behavior rather than surface token statistics.

Further, we have plenty of SD, self-SD, or SD-adjacent methods that focus on long context performance. Other than the above-discussed TriForce (Sun et al., 2024) and MagicDec (Sadhukhan et al., 2025), works like LongSpec (Yang et al., 2025), QuantSpec (Tiwari et al., 2025), and TokenSwift (Wu et al., 2025) also contribute. Their connection with FAFO mostly resides in the fact that FAFO is also evaluated on many long context tasks.

**Failure Modes of Lossy KV Cache Compression Methods**   Yuan et al. (2024) reveal that H2O (Zhang et al., 2023b) can perform decently on Needle-in-a-Haystack/passkey retrieval-like tasks (Mohtashami & Jaggi, 2023) if given a shorter passkey to retrieve and a continued prompt, but fails catastrophically (dropping from 100% to 35%) once the passkey length is extended. Similarly, benchmarks like SCBench (Li et al., 2025a) and later works like Ada-KV (Feng et al., 2024) reveal that while strong token dropping methods like SnapKV (Li et al., 2024a) are often performant across many tasks, they face significant performance degradation if they are unaware of the user query before eviction, thus directly hurting their multi-turn performance — arguably one of the signature capabilities of instruction-following LLMs. While this paragraph is in no way an exhaustive list of how lossy KV cache compressions can fail, it illustrates a recurring pattern worth noting: lossy KV cache compression introduces brittle, task-dependent weaknesses that are difficult to anticipate or detect without extensive, highly specific stress testing.

## C. Technical comparison of FAFO vs. Speculative Decoding methods

We first provide a detailed explanation of why FAFO can achieve a leveled memory footprint, while other SD baselines need to maintain additional KV cache. Recall that Speculative Decoding (SD) methods follow the sequential draft-THEN-verify pipeline. For such methods to be effective, the draft model must generate multiple (and consecutive) draft tokens; THEN, such draft tokens are verified.

Let us take the simplified TriForce (Sun et al., 2024) (ignoring the 68M tiny draft-draft model) as an example. From a fresh start (given a certain length of input, with no output yet), we have the following steps:

1. **Initial Prefill:** TriForce first identifies a set of token chunks from the prefill as "important" and evicts the rest, forming a lossy cache. At this point, this lossy cache is still a strict subset of the full/exact cache.

2. **Draft Token Generation:** TriForce then sequentially decodes multiple draft tokens, where the lossy cache naturally grows. Since the newly decoded draft tokens are generated upon the lossy KV, their own KV are also lossy. Thus, during draft generation, the lossy cache is no longer a subset of the full cache.

3. **Full Cache Verification & Lossy Cache Update:** After obtaining the draft tokens, TriForce engages in verification and obtains the full and exact cache of all accepted tokens. TriForce updates the lossy cache by replacing the accepted tokens' KV with the exact ones. It then evicts a number of "least important" tokens from the updated lossy cache to prevent its size from growing out of control.

4. **Repeat and Rebuild:** Steps 2 & 3 are repeated, and occasionally a full rebuild of the lossy cache is triggered, depending on various factors (e.g., low acceptance rate). TriForce cannot afford to materialize only one set of KV cache for two main reasons: a) During Step 2, its lossy and exact cache copies diverge, so some lossy KV must be stored; and b) Although the lossy cache becomes a subset of the full cache again by the end of Step 3 verification, generating draft tokens upon this lossy cache would require an updated and relatively fine-grained slicing/gathering (compared to something like a StreamingLLM-style masking) upon the full cache — which is a fairly costly operation to engage in at each verification step.

We note that this stands in contrast to n-gram candidate methods like FAFO, where verification occurs in parallel with the guess token generation. At each step, exactly one token is added per each "input" (# of input = # of n-grams + # verification + 1). In this setting, the lossy and full caches never diverge, so storing just one copy of the KV cache is sufficient. However, realizing efficiency gains under this setting still requires non-trivial efforts, as a naïve attention mask provides negligible

efficiency improvements over full attention and becomes a bottleneck in the parallel pipeline. To address this, we construct a static block-wise mask with FlexAttention and utilize our proposed swapping-based KV cache management design to maintain end-to-end efficiency.

# D. FAFO's tokens structure and attention mask

FAFO has both Fumble Around decoding and Find Out verification in a single forward pass of a single model, achieving a *draftless* setting: FAFO concatenates tokens from both the fumble decoding and verification phases, using a designated attention mask (Figure 4). In Fumble Around, instead of organizing the $n$ subsequences in a row-wise manner, we organize them in a column-wise manner (i.e., grouping tokens at the same position across subsequences step-by-step: $y_1^1, y_1^2, \ldots, y_1^n, y_2^1, y_2^2, \ldots, y_2^n, \ldots$), following the practice in (Fu et al., 2024). The attention mask of tokens in the fumble around decoding part is then constructed such that each token can only attend to tokens that (i) appear earlier in the sequence (i.e., have smaller position indices) and (ii) belong to the same subsequence (i.e., share the same column index). This column-wise organization significantly simplifies updating the sequence when new tokens are generated. Instead of shifting existing tokens and inserting new ones (as would be needed in a row-wise layout), FAFO only needs to *append* the newly generated tokens $y_{k+1}^1, y_{k+1}^2, \ldots, y_{k+1}^n$ to the end of the concatenated sequence. In contrast, tokens in the verification part follow the standard causal attention masking.

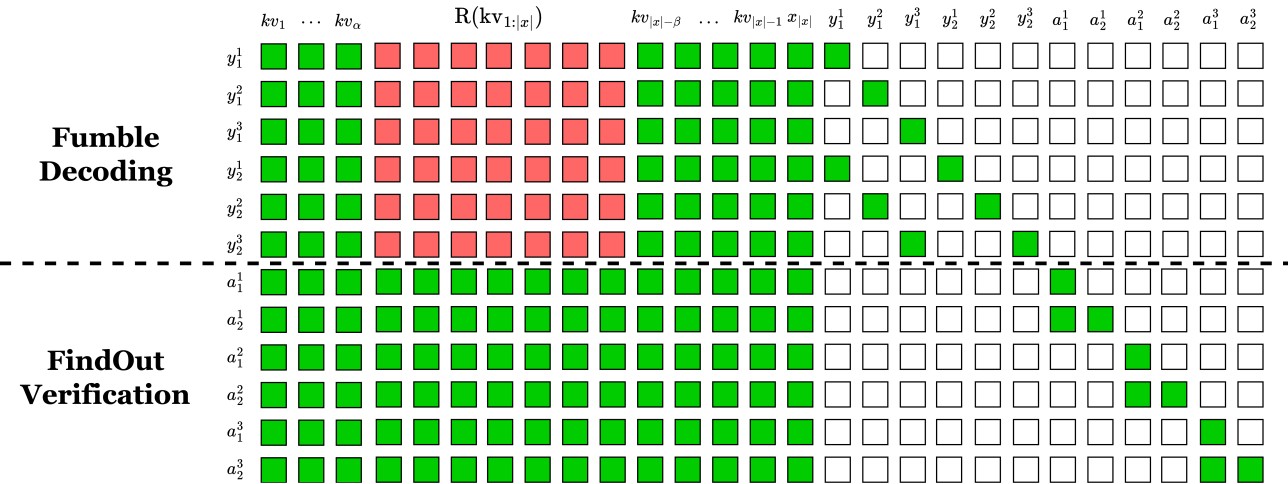

*Figure 4.* Attention mask for FAFO's concatenation of fumble around decoding and find out verification. Here, $x_{|x|}$ represents the latest token in the sequence while $kv_1 \ldots kv_{|x|-1}$ are the KV cache of previous tokens. Tokens $a_1^1, \ldots, a_2^2$ correspond to the verification phase and are flattened into a 1-D tensor with standard causal masking. Tokens $y_1^1, \ldots, y_2^2$ correspond to the fumble decoding phase and are flattened into a 1-D tensor using the designated masking described in Section 4.4 and Appendix D.

# E. Cache Pool Management

Although cached subsequences are offloaded to the CPU pool to save GPU memory, the pool size is expected to grow linearly, adding more load to the verification process later. On the other hand, subsequences generated earlier have a lower chance of being accepted later as the decoded sequence length increases more and more. To mitigate this, we limit the number of cached subsequences per starting token to $n$. Older candidate subsequences are evicted in a least-recently-used (LRU) manner.

We include the following proposition to provide principled intuition for why the prefix-based retrieval strategy works, offering a motivating explanation for an empirically effective design choice rather than a theoretical foundation for the method.

**Proposition E.1** (Monotonicity of verification success under longer matching prefixes). *Let $p_\theta$ be an autoregressive language model over sequences $y_{1:T}$,*

$$p_\theta(y_{1:T}) = \prod_{t=1}^{T} p_\theta(y_t \mid y_{1:t-1}).$$

*Fix a ground-truth sequence $y_{1:T}^\star$ and define, for any prefix length $\ell < T$,*

$$q_\ell := p_\theta(y_{\ell+1:T}^\star \mid y_{1:\ell}^\star) = p_\theta(y_{\ell+1}^\star, \ldots, y_T^\star \mid y_1^\star, \ldots, y_\ell^\star).$$

*Then for all $\ell < T - 1$,*

$$q_{\ell+1} \geq q_\ell.$$

*In other words, the probability that the model exactly reproduces the remaining suffix of $y^\star$ is non-decreasing in the length of the matching prefix.*

*Proof.* By the chain rule for $p_\theta$ and the definition of $q_\ell$,

$$q_\ell = p_\theta\left(y^\star_{\ell+1:T} \,\middle|\, y^\star_{1:\ell}\right) = p_\theta\left(y^\star_{\ell+1} \,\middle|\, y^\star_{1:\ell}\right) p_\theta\left(y^\star_{\ell+2:T} \,\middle|\, y^\star_{1:\ell+1}\right).$$

The second factor on the right-hand side is exactly $q_{\ell+1}$:

$$q_{\ell+1} = p_\theta\left(y^\star_{\ell+2:T} \,\middle|\, y^\star_{1:\ell+1}\right).$$

Hence

$$q_\ell = p_\theta\left(y^\star_{\ell+1} \,\middle|\, y^\star_{1:\ell}\right) q_{\ell+1}.$$

Since $p_\theta\left(y^\star_{\ell+1} \,\middle|\, y^\star_{1:\ell}\right) \in (0, 1]$, we obtain

$$q_{\ell+1} = \frac{q_\ell}{p_\theta\left(y^\star_{\ell+1} \,\middle|\, y^\star_{1:\ell}\right)} \geq q_\ell.$$

Thus $q_{\ell+1} \geq q_\ell$ for all $\ell < T - 1$, which proves the claim. $\square$

# F. Additional Experiments

Due to page limitations, we can only include so many experiments in the main text. Here, we present additional experiment results on FAFO under a non-greedy decoding setting, its performance under reasoning tasks (as reasoning models tend to decode a lot of chain-of-thought tokens before delivering the final answer, making up a huge presence in KV cache), and how it holds up against SCBench[4], which is often regarded as one of the hardest benchmarks for lossy KV cache compression methods. Last, we conduct ablation studies on hyper-parameters crucial to FAFO.

## F.1. Sampling Decoding

Table 4 presents FAFO vs Lookahead under a non-greedy sampling setting. For each model, we adopt its default `generation_config.json` for the sampling setting, as different models might prefer a different sampling setup. Specifically, we have:

- **Llama-2-7b-chat**: temperature $T = 0.6$, top-p $= 0.9$

- **Llama-3-8B-Instruct**: temperature $T = 0.6$, top-p $= 0.9$

- **Llama-3.1-8B-Instruct**: temperature $T = 0.6$, top-p $= 0.9$

- **Qwen2.5-7B-Instruct**: temperature $T = 0.7$, top-p $= 0.8$, top-k $= 20$

- **Qwen2.5-32B-Instruct**: temperature $T = 0.7$, top-p $= 0.8$, top-k $= 20$

*Table 4.* Observed practical latency speedup and average acceptance length $\tau$ on MT-bench, GSM8K, and HumanEval-Completion with sampling temperature $T > 0$.

| Models | Method | MT-bench | | GSM8K | | HumanEval-C | |
|---|---|---|---|---|---|---|---|
| | | Speedup | $\tau$ | Speedup | $\tau$ | Speedup | $\tau$ |
| **Llama-2-7b-chat** | FAFO-Stream | **1.91**× | **2.25** | **1.95**× | **2.72** | **1.87**× | **2.35** |
| | FAFO-Quest | 1.78× | 2.19 | 1.71× | 2.60 | 1.87 × | 2.34 |
| | Lookahead | 1.64× | 2.01 | 1.68× | 2.50 | 1.79× | 2.33 |
| **Llama-3-8B-Instruct** | FAFO-Stream | **1.58**× | **2.06** | **1.57**× | **2.09** | **1.58**× | **2.00** |
| | FAFO-Quest | 1.47× | 2.00 | 1.46× | 2.02 | 1.49× | 2.01 |
| | Lookahead | 1.40× | 1.99 | 1.43× | 2.03 | 1.50× | 1.98 |
| **Llama-3.1-8B-Instruct** | FAFO-Stream | **1.39**× | 2.08 | **1.41**× | **2.10** | **1.58**× | **2.22** |
| | FAFO-Quest | 1.37× | **2.10** | 1.36× | 2.07 | 1.50× | 2.20 |
| | Lookahead | 1.28× | 2.00 | 1.28× | 2.03 | 1.45× | 2.08 |
| **Qwen2.5-7B-Instruct** | FAFO-Stream | **1.48**× | **2.16** | **1.56**× | **2.30** | **1.55** × | **2.10** |
| | FAFO-Quest | 1.41× | 2.03 | 1.48× | 2.26 | 1.41× | 2.07 |
| **Qwen2.5-32B-Instruct** | FAFO-Stream | **1.25**× | 2.10 | **1.35**× | **2.40** | **1.44**× | **2.35** |
| | FAFO-Quest | 1.12× | 2.02 | 1.23× | 2.32 | 1.33× | **2.35** |
| | Lookahead | 1.20× | **2.12** | 1.16× | 2.35 | 1.27× | 2.30 |

## F.2. Robustness under Reasoning-Intensive Task

Table 5 reports speedup and average acceptance length $\tau$ on AIME24, a challenging mathematical reasoning benchmark. Across both distilled backbones, FAFO-Stream achieves the highest speedup compared to FAFO-Quest and Lookahead, while maintaining comparable or slightly higher acceptance lengths. The consistent improvement in speedup without a corresponding inflation in $\tau$ suggests that FAFO-Stream benefits primarily from reduced verification overhead rather than

---

[4]https://huggingface.co/datasets/microsoft/SCBench

more aggressive speculative acceptance. Notably, the relative ordering of methods is preserved across model families, indicating that FAFO's gains are not model-specific but instead stem from its decoding design.

*Table 5.* Speedup ratio and average acceptance length $\tau$ on AIME24.

| Models | Method | Speedup | $\tau$ |
|---|---|---|---|
| **DeepSeek-R1-Distill-Qwen-7B** | FAFO-Stream | **1.60**$\times$ | 2.40 |
| | FAFO-Quest | 1.46$\times$ | 2.35 |
| | Lookahead | 1.37$\times$ | 2.26 |
| **DeepSeek-R1-Distill-Llama-8B** | FAFO-Stream | **1.65**$\times$ | 2.41 |
| | FAFO-Quest | 1.50$\times$ | 2.27 |
| | Lookahead | 1.48$\times$ | 2.40 |

## F.3. Robustness under Multi-turn Evaluation (Multi-IF)

Table 6 evaluates FAFO variants on Multi-IF, which emphasizes multi-step inference and longer reasoning chains. FAFO-Stream again outperforms FAFO-Quest and Lookahead, achieving over 2$\times$ speedup while sustaining a high acceptance length. Compared to AIME24, Multi-IF yields larger $\tau$ values across all methods, reflecting the increased predictability of longer reasoning segments. Within this regime, FAFO-Stream translates higher acceptance into tangible throughput gains more effectively than competing approaches, highlighting its robustness on multi-hop inference workloads.

*Table 6.* Speedup ratio and average acceptance length $\tau$ on Multi-IF.

| Models | Method | Speedup | $\tau$ |
|---|---|---|---|
| **Llama-2-7b-chat** | FAFO-Stream | **2.02**$\times$ | 2.84 |
| | FAFO-Quest | 1.76$\times$ | 2.76 |
| | Lookahead | 1.69$\times$ | 2.73 |

## F.4. Robustness under SCBench

Table 7 presents results on SCBench, which spans diverse task categories including mathematical problem solving, repository-level QA, in-context learning with many shots, and multi-hop retrieval. FAFO-Stream delivers consistent speedups across all subsets, with acceptance lengths remaining stable around $\tau \approx 2.6$–2.9. The relatively uniform performance across heterogeneous tasks suggests that FAFO does not rely on task-specific heuristics or prompt structure, and that its draftless streaming mechanism generalizes well to mixed workloads encountered in practical inference settings.

*Table 7.* Speedup ratio and average acceptance length $\tau$ on datasets from SCBench.

| Models | Method | Math.Find | | RepoQA | | ICL.ManyShot | | Retr.MultiHop | |
|---|---|---|---|---|---|---|---|---|---|
| | | Speedup | $\tau$ | Speedup | $\tau$ | Speedup | $\tau$ | Speedup | $\tau$ |
| **L3.1 8B** | FAFO-Stream | 1.37$\times$ | 2.88 | 1.26$\times$ | 2.60 | 1.26$\times$ | 2.61 | 1.46$\times$ | 2.90 |

## F.5. More KV Cache Compression Method

We feature a third KV-cache compression method — SnapKV — in addition to Stream and Quest, which we have extensively benchmarked. This complementary design broadens the applicability of our approach across tasks and models.

## F.6. FAFO vs. TriForce

Because TriForce's native implementation is incompatible with GQA, and also to ensure that we are not running it in a much disadvantaged setting (as the 0.21$\times$ seems abnormal from a quick scan), we tested TriForce in its own reported setting (Yarn-Llama-2-7b-128k with PG-19 and NarrativeQA) in Table 10 and Table 9. In this setting, we do observe significant

*Table 8.* Speedup ratio and average acceptance length $\tau$ on **Llama3-8B-Instruct** with **FAFO-SnapKV**. Entries show *speedup*× ($\tau$).

| Models | Method | MTBench | HumanEval | Multi-IF |
|---|---|---|---|---|
| **Llama3-8B-Instruct** | FAFO-SnapKV | 1.69× (1.90) | 1.87× (2.41) | 1.81× (2.47) |

practical speedup from TriForce (though still below FAFO's) and 5×+ of $\tau$; this hints at two conclusions: 1) TriForce is most performant under easy general language modeling tasks, but not challenging, goal-specific tasks; and 2) even under such tasks, TriForce's implementation bottlenecks prevent it from fully leveraging its high guess generation quality, likely because of the complexity of its three-model pipeline. In contrast, FAFO is capable of offering decent improvement over all reported tasks.

*Table 9.* Speedup ratio and average acceptance length $\tau$ on **TriForce's NarrativeQA** with **Yarn-Llama-2-7b-128k**. Entries show *speedup*× ($\tau$).

| Model | Method / Input Length | 3072 | 5120 | 10240 |
|---|---|---|---|---|
| **Yarn-Llama-2-7b-128k** | TriForce | 1.44× (4.26) | 1.37× (4.12) | 0.29× (0.11) |
| | FAFO-Stream | **2.50**× (4.23) | **1.80**× (4.01) | **1.24**× (4.07) |

*Table 10.* Speedup ratio and average acceptance length $\tau$ on different context lengths of PG-19.

| Models | Method | 1024 | | 2048 | | 3072 | |
|---|---|---|---|---|---|---|---|
| | | Speedup | $\tau$ | Speedup | $\tau$ | Speedup | $\tau$ |
| **Yarn-Llama-2-7b-128k** | FAFO-Stream | **2.71**× | 3.03 | **1.90**× | 2.84 | **1.80**× | 2.65 |
| | FAFO-Quest | 2.03× | 2.38 | 1.35× | 2.10 | 1.22× | 1.93 |
| | Lookahead | 1.75× | 2.53 | 1.36× | 2.51 | 1.16× | 2.55 |
| | TriForce | 1.83× | 5.75 | 1.63× | 5.14 | 1.74× | 5.50 |

### F.7. Additional Baselines

We provide head-to-head comparisons against two additional baselines: ANPD (Ou et al., 2024) and SparseSpec (Zhao et al., 2025). All results below are measured on Llama-3.1-8B-Instruct with the same evaluation setup as Table 2.

**FAFO vs. ANPD.** ANPD is a draftless lossless n-gram decoder that draws draft tokens from a token-level statistical (Markov) n-gram model built during a warm-up phase, then verifies via the standard sequential draft-then-verify pipeline. In contrast, FAFO generates guesses by running the target model itself under a compressed KV cache and maintains a dynamic token pool that continuously updates from the ongoing context. We report ANPD throughput excluding its n-gram warm-up phase, giving it the most favorable measurement setting. As shown in Table 11, FAFO-Stream outperforms both ANPD and SWIFT across all benchmarks.

*Table 11.* Llama-3.1-8B-Instruct: FAFO vs. ANPD and SWIFT. ANPD throughput excludes its n-gram warm-up phase.

| Method | MT-Bench | | GSM8K | | HumanEval-C | |
|---|---|---|---|---|---|---|
| | Speedup | $\tau$ | Speedup | $\tau$ | Speedup | $\tau$ |
| FAFO-Stream | **1.50**× | 2.10 | **1.31**× | 2.08 | **1.57**× | 2.20 |
| SWIFT | 0.94× | 1.53 | 0.98× | 3.22 | 1.03× | 2.38 |
| ANPD | 0.87× | 1.53 | 0.52× | 1.52 | 0.78× | 1.60 |

**FAFO vs. SparseSpec.** SparseSpec (Zhao et al., 2025) is a concurrent work that similarly pairs self-speculation with KV cache compression but follows the standard sequential draft-then-verify pipeline. SparseSpec achieves higher raw speedups on most benchmarks; however, we argue that this gap is primarily attributable to system-level engineering advantages that SparseSpec has accumulated: CUDA graph capture with double buffering, fused draft+verify FlashInfer (Ye et al., 2025) kernels, asynchronous CPU scheduling, and KV cache offloading. To validate this, we integrated FlashInfer kernel

support into FAFO per reviewer suggestion, as shown in Table 12. With only a simple system-level support added, FAFO (FlashInfer) substantially closes the gap on GSM8K and HumanEval, confirming that the remaining throughput difference reflects ongoing system engineering work orthogonal to the core research contribution.

*Table 12.* Llama-3.1-8B-Instruct: FAFO vs. SparseSpec. FAFO (FlashInfer) integrates FlashInfer's single-sequence prefill kernel to encode FAFO's access pattern without the fixed block-size constraint of FlexAttention.

| Method | GSM8K Speedup | HumanEval Speedup | MT-Bench Speedup |
|---|---|---|---|
| SparseSpec | 2.15× | 2.73× | 2.47× |
| FAFO (FlexAttn) | 1.31× | 1.57× | 1.50× |
| FAFO (FlashInfer) | 2.07× | 2.10× | 1.45× |

We additionally show that SparseSpec's PillarAttention compression kernel is integrable into FAFO's fumble decoding as a drop-in KV cache compression backend (Table 13), demonstrating FAFO's compatibility with diverse compression methods.

*Table 13.* Llama-3.1-8B-Instruct: FAFO with PillarAttention as the KV cache compression backend, demonstrating compatibility with SparseSpec's compression kernel.

| Benchmark | Speedup vs. Dense | Compression Ratio |
|---|---|---|
| MT-Bench | 1.18× | 2.58× |
| HumanEval-C | 1.11× | 2.65× |

**FAFO with FlashInfer.** We integrate FlashInfer's (Ye et al., 2025) single-sequence prefill kernel with a precomputed, token-level boolean attention mask that encodes the exact FAFO access pattern (sink tokens + sliding window + speculative frame). This achieves token-level mask granularity without the fixed block-size constraint of FlexAttention's Triton backend, and without any paging overhead. As shown in Table 14, FAFO-FlashInfer achieves substantially higher speedup on GSM8K and HumanEval, while slightly underperforming on MT-Bench. We attribute this to task characteristics: mathematical reasoning and code generation exhibit highly structured, repetitive token patterns that yield strong n-gram speculation candidates, whereas open-ended multi-turn dialogue is more lexically diverse — reducing the density of reusable n-gram chains and leading to a modest reduction in speculative acceptance rate.

*Table 14.* Llama-3.1-8B-Instruct: FAFO with FlexAttention vs. FlashInfer kernel backend.

| Method | MT-Bench | | GSM8K | | HumanEval-C | |
|---|---|---|---|---|---|---|
| | Speedup | $\tau$ | Speedup | $\tau$ | Speedup | $\tau$ |
| FAFO (FlexAttn) | **1.50×** | 2.10 | 1.31× | 2.08 | 1.57× | 2.20 |
| FAFO (FlashInfer) | 1.45× | 2.54 | **2.07×** | 3.02 | **2.10×** | 2.31 |

### F.8. Ablation Study

We conduct an ablation study on different numbers of guesses and on the length of each $k$-gram guess. On MT-Bench (Table 15), FAFO-Stream exhibits a clear interior optimum: short guesses ($k=4$) underperform ($0.90×$, $\tau=1.90$), while mid-range guesses around $k\approx6$–$7$ yield the highest speedups ($\sim1.9×$) with only modest increases in acceptance length ($\tau\approx2.10$–$2.14$). Pushing $k$ beyond this range slightly tapers speedup (e.g., $k=8$–$9$: $1.75$–$1.81×$) without $\tau$ benefits, indicating diminishing returns once guesses get too long.

*Table 15.* Speedup ratio and average acceptance length $\tau$ on different lengths of $k-gram$ guess subsequences on MT-Bench.

| Models | Method | 4 | | 5 | | 6 | | 7 | | 8 | | 9 | |
|---|---|---|---|---|---|---|---|---|---|---|---|---|---|
| | | Speedup | $\tau$ | Speedup | $\tau$ | Speedup | $\tau$ | Speedup | $\tau$ | Speedup | $\tau$ | Speedup | $\tau$ |
| **Llama-2-7b-chat** | FAFO-Stream | 0.90× | 1.90 | 1.03× | 2.04 | 1.90× | 2.10 | 1.90× | 2.14 | 1.75× | 2.20 | 1.81× | 2.15 |

Varying the number of parallel guess subsequences (Table 16) consistently increases speedup: FAFO-Stream improves from $\sim1.05×$ at 10 to $\sim1.92×$ at 40, and $\tau$ rises smoothly from $\sim1.9$ to $\sim2.3$. FAFO-Quest follows the same monotonic trend.

*Table 16.* Speedup ratio and average acceptance length $\tau$ on different number of guess subsequences.

| Models | Method | 10 | | 20 | | 30 | | 40 | |
|---|---|---|---|---|---|---|---|---|---|
| | | Speedup | $\tau$ | Speedup | $\tau$ | Speedup | $\tau$ | Speedup | $\tau$ |
| **Llama-2-7b-chat** | FAFO-Stream | $1.05\times$ | 1.92 | $1.78\times$ | 2.10 | $1.81\times$ | 2.20 | $1.92\times$ | 2.30 |
| | FAFO-Quest | $0.90\times$ | $1.86\times$ | $1.59\times$ | 2.10 | $1.64\times$ | 2.17 | $1.70\times$ | 2.19 |

Finally, we conduct a compression ratio ablation study (Table 17) on Multi-IF with Llama-2-7b-chat. Results show a familiar trade-off: a moderate Init+Local token budget ($\approx 760$ tokens) maximizes speedup ($2.02\times$) while keeping $\tau$ flat ($\approx 2.84$), whereas overly aggressive compression (1360) lowers speedup ($1.59\times$).

*Table 17.* Speedup ratio and average acceptance length $\tau$ on **Multi-IF** with **Llama-2-7b-chat** across different compression settings. Entries show *speedup*$\times$ ($\tau$).

| Model | Method / Init+Local Tokens | 360 | 560 | 760 | 1360 |
|---|---|---|---|---|---|
| **Llama-2-7b-chat** | FAFO-Stream | $1.93\times$ (2.76) | $1.98\times$ (2.83) | **2.02**$\times$ (2.84) | $1.59\times$ (2.86) |

Table 18 examines the sensitivity of FAFO to the lookback window length. Speedup remains stable across a wide range of window sizes, with only marginal variation in both throughput and $\tau$. This robustness suggests that FAFO does not rely on finely tuned temporal heuristics, and that its efficiency is driven by aggregate acceptance behavior rather than precise window calibration.

*Table 18.* Different lookback window lengths with Llama-2-7b-chat on MT-Bench.

| lookback window length | Speedup | $\tau$ |
|:---:|:---:|:---:|
| 3 | 1.89x | 2.28 |
| 5 | 1.94x | 2.29 |
| 7 | 1.97x | 2.29 |
| 10 | 1.92x | 2.28 |

## G. KV Cache Management

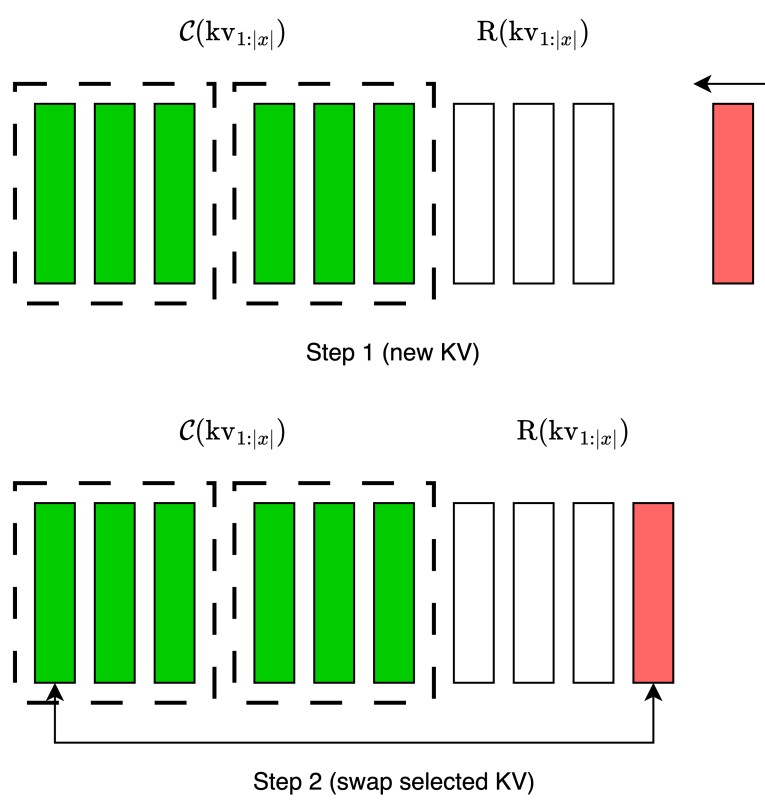

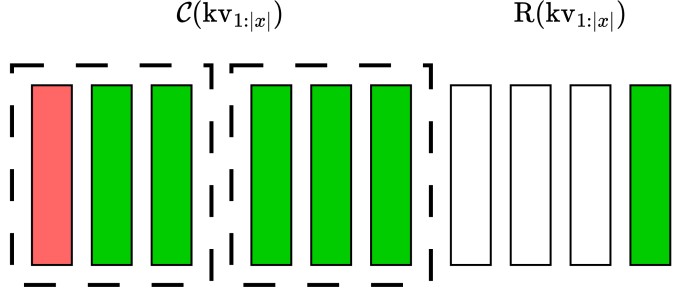

*Figure 5.* KV Cache Management with selective swapping into fixed-size KV blocks. The red KV cache represents the newly generated entry at the current decoding step, while the green KV caches denote previously selected entries retained by the compression function $\mathcal{C}$.

In Figure 5, suppose we are using StreamingLLM as the KV cache compression function. After each decoding step, new KV cache entries are generated (Step 1). FAFO manages the cache by swapping the new KV entries with discarded ones — the oldest ones in this case, utilizing StreamingLLM as function $\mathcal{C}$. This operation takes place within a fixed region of KV cache blocks that is shared by both the Fumble Decoding and Find Out phases, ensuring that all relevant KV entries remain compactly located.

This design is well-suited to FlexAttention, which improves efficiency by skipping over blocks that are entirely sparse (Dong et al., 2025). Since FAFO maintains the active KV entries within a fixed number of blocks (two blocks in Figure 5), FlexAttention only needs to load these specific blocks into the GPU's streaming multiprocessors (SMs) for attention computation. In contrast, without FAFO, the KV cache becomes scattered across all cache blocks, forcing FlexAttention to load the entire KV cache, even when most blocks contain only discarded or irrelevant entries. A visual comparison is shown in Figure 3 of the main text.

Moreover, we would like to emphasize that the *cost of precomputing the BlockMask for FlexAttention is prohibitively high*, rendering it impractical in the context of the dynamic and ever-growing KV cache during the autoregressive inference process. While one might argue that in methods such as STREAMLLM, where the number of KV cache blocks remains constant, FLEXATTENTION could similarly load only a small number of blocks, this overlooks a critical limitation: maintaining this efficiency would require *recomputing the BlockMask at every decoding step* to reflect the current structure of the compressed KV cache. Unfortunately, this recomputation is **extremely expensive — often more costly than simply loading the full KV cache into memory**. As a result, without a mechanism like FAFO to ensure locality and block consistency over time, the theoretical benefits of sparse attention are outweighed by the overhead of managing the sparsity structure itself.

FAFO implements a *draftless* pipeline by fusing the Fumble Around (speculation) and Find Out (verification) phases into a single model forward pass. By concatenating draft tokens and verification tokens, we leverage a unified attention mask (Figure 4) to maximize GPU utilization. However, simply masking tokens logically does not guarantee system efficiency.

### G.1. System-Efficient FAFO via Sparse Attention Kernels

**The Memory Wall in Sparse Attention.** While the logical attention matrix for FAFO is highly sparse, standard attention kernels are agnostic to this structure. They typically load the entire KV cache from High Bandwidth Memory (HBM) into on-chip SRAM to compute attention scores, applying masks only *after* the expensive memory load operations. Consequently, the decoding latency remains **memory-bandwidth bound**, scaling linearly with the full context length $|x|$ rather than the sparse subset size. To overcome this, we employ FlexAttention to enforce block-sparse computation, but its naive application introduces new overheads.

**The Challenge of Dynamic Masking.** While FlexAttention is well-suited for the prefill phase, where the input length remains static, applying it to the decoding phase is highly non-trivial. The context grows with each generated token, requiring the block mask to be recomputed at every step. **This recomputation is prohibitively costly, rendering naive use of FlexAttention during decoding impractical** without a decoding-aware block masking scheme specifically designed to handle dynamic attention contexts. Specifically, dynamic mask regeneration triggers *repeated graph capture and compilation overheads*, effectively negating the computational benefits of sparsity by stalling the GPU pipeline.

**Hardware-Aware KV Layout for Zero-Overhead Masking.** We resolve this by decoupling the *logical* position of tokens from their *physical* memory location. We implement a **Physically Contiguous, Logically Sparse** memory management strategy. We allocate a fixed buffer of physical KV blocks, $\mathbf{B}_{fixed}$, at the head of the KV tensor to store the compressed entries $\mathcal{C}(\text{kv}_{1:|x|})$.

$$\text{KV Cache Physical Layout} = \left[ \underbrace{\mathcal{C}(\text{kv}_{1:|x|})}_{\text{Selected Blocks}} \ \| \ \underbrace{\text{R}(\text{kv}_{1:|x|})}_{\text{Irrelevant Blocks}} \right] \tag{2}$$

By enforcing that all relevant tokens for the current "Fumble Around" step reside within $\mathbf{B}_{fixed}$, we construct a static FlexAttention block mask that instructs the kernel to load *only* the first $|\mathbf{B}_{fixed}|$ blocks from HBM to SRAM. Irrelevant entries $\text{R}(\text{kv})$ are logically preserved but physically skipped during the attention kernel execution (Algorithm 1).

---

**Algorithm 1** FAFO-Optimized FlexAttention Kernel (Block-Parallel)

---

**Require:** Query $Q$, Key $K$, Value $V$ in HBM
**Require:** Fixed Block Budget $N_{active}$, Block Size $B_z$
**Require:** Grid Dimension $M = \lceil \text{seq\_len}/B_z \rceil$
 1: **// Kernel Grid Launch: Each Thread Block handles one Query Tile**
 2: **for all** Query Block index $i \in \{0, \dots, M-1\}$ **in parallel do**
 3:    Load $Q_i = Q[i \cdot B_z : (i+1) \cdot B_z]$ from HBM $\rightarrow$ SRAM
 4:    Initialize accumulator $O_i \leftarrow 0$
 5:    **for** $j = 0$ **to** $N_{active} - 1$ **do**
 6:       **// Load Key/Value Blocks**
 7:       Load $K_j, V_j$ from HBM address $[j \cdot B_z] \rightarrow$ SRAM
 8:       **// Block-wise Attention Computation**
 9:       $S_{ij} \leftarrow Q_i \cdot K_j^T$
10:       Apply logical score mod: $S_{ij} \leftarrow S_{ij} + M_{logic}[i, j]$
11:       **// Update Output Accumulator**
12:       $O_i \leftarrow \text{Update}(O_i, S_{ij}, V_j)$
13:    **end for**
14:    **// 3. Writeback Result**
15:    Write $O_i$ from SRAM $\rightarrow$ HBM address $[i \cdot B_z]$
16: **end for**
17: **Result:** Memory traffic reduced to $O(M \cdot N_{active} \cdot B_z)$

---

## H. Verification on Lossless Generation Quality of FAFO

As theoretically proven in (Leviathan et al., 2023), FAFO preserves lossless generation quality. Following Lookahead Decoding's evaluation practice, we benchmark on LLaMA-2-7B-Chat over 160 samples from MT-Bench using Hugging Face greedy as the reference. Under FP32, FAFO reproduces the greedy outputs exactly on 157/160 samples, whereas the remaining cases differ by only 3–10 characters. Under FP16, FAFO's outputs align with the corresponding Hugging Face greedy results, while exhibiting small deviations from the FP32 reference (36/160). Thus, although practical runs at reduced precision may not perfectly match Hugging Face greedy, FAFO retains the greedy output distribution within the numerical error range — no worse than Hugging Face's half-precision behavior — while remaining lossless by construction under identical numerical settings.

*Table 19.* Full GPU vs. CPU-offloading with Llama-2-7b-chat on MT-Bench.

| Baselines | Speedup | $\tau$ |
|---|---|---|
| FAFO (GPU) | 1.36x | 2.36 |
| FAFO (CPU-offloading) | 1.91x | 2.29 |
| Lookahead (GPU) | 1.27x | 1.66 |
| Lookahead (CPU-offloading) | 1.61x | 1.66 |

*Table 20.* Full GPU vs. CPU-offloading with Llama-2-7b-chat on HumanEval.

| Baselines | Speedup | $\tau$ |
|---|---|---|
| FAFO (full GPU) | 1.34x | 2.41 |
| FAFO (CPU-offloading) | 2.03x | 2.34 |
| Lookahead (full GPU) | 1.23x | 1.82 |
| Lookahead (CPU-offloading) | 1.72x | 1.77 |

## I. GPU–CPU Offloading, Compute Efficiency, and Design Trade-offs

Tables 19 and 20 study the impact of CPU offloading under identical model and decoding configurations. Across both MT-Bench and HumanEval, CPU offloading consistently improves end-to-end speedup for speculative decoding methods, with FAFO benefiting more substantially than Lookahead. In particular, FAFO gains an additional 0.5–0.7$\times$ speedup under offloading, while maintaining comparable acceptance thresholds $\tau$. This suggests that FAFO's draft generation and verification pipeline is less sensitive to host–device transfer overheads and better amortizes CPU-side work through higher verification efficiency.

Table 21 highlights the trade-off between computational cost and throughput as the number of generated guesses per step increases. While increasing guesses raises TFLOPs per step almost linearly, speedup exhibits a non-monotonic behavior: performance improves from 10 to 20 guesses, but degrades at 30 guesses. This indicates diminishing returns from aggressive speculation, where additional draft computation outweighs verification savings. In practice, a moderate number of guesses provides the best balance between compute overhead and decoding acceleration.

Table 22 compares FAFO with Lookahead and vanilla decoding under a unified compute-efficiency lens. Although both speculative methods incur substantially higher TFLOPs per step than vanilla decoding, their effective speedups are constrained by implementation overheads, as reflected in the gap between idealized $\tau$-based speedup and realized throughput. Notably, FAFO-Stream achieves a comparable overhead penalty to Lookahead despite a simpler, draftless design, indicating that its performance gains primarily stem from improved verification efficiency rather than increased speculation depth.

## J. "Find Out" Caching, Retrieval, and Verification Algorithm

---

**Algorithm 2** "Find Out" Caching

---

**Input:** A cache pool $G$, $n$ subsequences $y^1_{s_1+2:s_1+k+1}, \cdots, y^n_{s_n+2:s_n+k+1}$, and their buffers of discarded tokens $y^1_{s_1-k:s_1+1}, \cdots, y^n_{s_n-k:s_n+1}$
**Output:** Updated cache pool $G$
**for** $i = 1$ **to** $n$ **do**
    **for** $j = s_1 + 1$ **down to** $s_1 - k$ **do**
        $G[(y^1_{j:s_1+1})]$.add($y^i_{s_i+2:s_i+k+1}$)
    **end for**
**end for**
**return** $G$

---

*Table 21.* TFLOPs/step and speedup for different numbers of generated guesses per step of Llama-2-7b-chat on MT-Bench.

| # guesses | TFLOPs/step | Speedup |
|-----------|-------------|---------|
| 10        | 2.43        | 1.69x   |
| 20        | 3.38        | 1.89x   |
| 30        | 4.44        | 1.75x   |

*Table 22.* Comparison with Lookahead and vanilla full-model decoding for Llama-2-7b-chat on MT-Bench.

| Method      | TFLOPs/step | $\tau$ - Speedup (implementation overhead) |
|-------------|-------------|--------------------------------------------|
| Vanilla     | 0.02        | $0\times$ (no overhead)                    |
| Lookahead   | 3.63        | $(2.01\text{–}1.64)=0.37\times$            |
| FAFO-Stream | 3.38        | $(2.25\text{–}1.91)=0.34\times$            |

---

**Algorithm 3** "Find Out" Retrieval

---

**Input:** An input sequence $x_{1:|x|}$, a cache pool $G$, number of sequence to be retrieved for verification $m$, sequence length $k$
**Output:** $m$ retrieved sequences for verification
$\{\{\}$ denotes a set$\}$
retrievedSeqs $\leftarrow \{\}$
suffixSeqLen $\leftarrow k + 2$

**while** size(retrievedSeqs) $< m$ & suffixSeqLen $> 0$ **do**
   **for** sequence $y \in G[x_{|x|-\text{suffixSeqLen}:|x|}]$ **do**
      retrievedSeqs $\leftarrow$ retrievedSeqs $\cup y$
      **if** size(retrievedSeqs)$= m$ **then**
         **Break**
      **end if**
   **end for**
**end while**
**return** retrievedSeqs

---

---

**Algorithm 4** "Find Out" Verification

---

1: **Input:** An input sequence $x_{1:|x|}$, a language model $p$, $m$ candidate sequences $A = \{a_{1:k}^1, \ldots, a_{1:k}^m\}$
2: **Output:** accepted tokens $c$
3: $c \leftarrow \emptyset$
4: **for** $a \in A$ **do**
5:     $\mathcal{D} \leftarrow p(x_{|x|}|x_{1:|x|-1})$
6:     $c' \leftarrow \emptyset$
7:     **for** $i = 1$ **to** $n$ **do**
8:         **if** $\text{argmax}(\mathcal{D}) = a_i$ **then**
9:             $\mathcal{D} \leftarrow p(a_i|c', x_{1:|x|})$
10:            $c' \leftarrow [c'|a_i]$
11:         **else**
12:            **break**
13:         **end if**
14:     **end for**
15:     $c' \leftarrow [c'|\text{argmax}(\mathcal{D})]$
16:     **if** $\text{size}(c') > \text{size}(c)$ **then**
17:         $c \leftarrow c'$
18:     **end if**
19: **end for**
20: **return** $c$

---

