# OpenReview forum: "FAFO: Lossy KV Cache Compression for Lossless Inference Acceleration via Draftless Fumble Decoding"
_ICML.cc/2026/Conference — ICML 2026 regular_

### Official Review · Reviewer_5Qf5 · 2026-03-09

**Soundness:** 2
**Presentation:** 2
**Significance:** 3
**Originality:** 2
**Overall Recommendation:** 3
**Confidence:** 3

**Summary:**

Previous lossy efficiency methods do fail under certain models or tasks setting. And when they fail they often fail catastrophically, which makes the real-world application not feasible. In this work, they present a framework that can be regarded as lossless generation quality while using lossy KV cache compression which is named as Fumble Around and Find Out (FAFO). It leverages a modified version of the n-gram candidate-pool decoding paradigm. This method does not save any memory but improves the latency of during inference.  They claim that their framework is compatible with all typical static or dynamic KV cache compression methods. The basic idea is that they use a compressed KV cache model to generate multiple n-gram and use the full kv-cache model to verify the generation, allowing multiple tokens to be accepted at once while preserving lossless generation quality.

**Compliance With Llm Reviewing Policy:**

Affirmed.

**Key Questions For Authors:**

What is the main contribution beyond combining existing methods (speculative decoding + n-gram + kv-compression?


Do you report the total required FLOPs compared to vanilla decoding?


How to choose the optimal number of guesses?

**Limitations:**

yes

**Strengths And Weaknesses:**

Strengths:


The submission is technically sound and claims in the paper are well-supported. The idea is reasonable and sound. The proposed method takes advantage of the idea and insights from speculative decoding and n-gram parallel guess. The relation with previous works are discussed thoroughly.




Weakness:


The writing and presentation can be improved:
For example, the font size in each table is very different. For example,table 6 and table 7.
There are too many bold fonts in the text such as line 200, line 209.


It is good to have detailed comparison with previous work but it takes too much space in your main pages. In the introduction, you should just introduce the novelty of your method and briefly discuss the downsides of previous methods.


Line 252, why is there a footnote external link to another blog? You should just explain their GIF idea and cite it.


‘Link 241: we expect many in our audience to be unfamiliar with the implementation details of Lookahead Decoding.’ ->this sentence is unnecessary to write in the paper


‘Line 248: it is our honest assessment’ no need to mention this.




n-gram decoding lacks effective support for batched inference


How sensitive is FAFO to compression ratio? Do you choose different compression ratio for different tasks?

---

> ### Author Rebuttal · Authors · 2026-03-31
>
> ### **`W1 - Presentation issues`**
> **On the bolding.** we fully acknowledge this point, though please excuse us to be candid: our use of bold is partly a deliberate accessibility choice, as prior reviewer experience has shown that key insights are easily missed due to highly technical nature of the paper to people not from SD background. We will revisit and reduce bolding where it feels excessive.
>
> **On introduction length and prior work discussion.** We appreciate this feedback, but want to be transparent about a genuine tension we face:
> - Our detailed prior work discussion was added specifically in response to past reviewers who were unfamiliar with the draftless decoding landscape and questioned our motivation — feedback that is in direct conflict with yours. This illustrates a recurring challenge: *presentation is often a multifaceted subject with no universally good answer*. Reviewers more familiar with the frontier naturally prefer a leaner introduction, while those coming from other areas benefit from the additional context. Indeed, while you prefer a shorter introduction, this preference is in clear contrast to:
> 	- R`BTdw` feedback, where *"Lookahead is not well explained within the paper"*.
> 	- R `mHiU` in contrast praises *"Clear presentation: The paper clearly contrasts FAFO with prior speculative decoding methods "*.
> - We ask for leniency when encountering background that feels familiar, as the general community or experts of different focuses may find such grounding genuinely useful. That said, we will do our best to restructure the introduction to front-load FAFO's novelty more clearly.
>
> **On the external blog footnote .** We believe Lookahead Decoding's original manuscript is not written well. Fully re-explaining it within our page budget would crowd out FAFO's own content. We have instead tried to provide the most useful resources available: we strongly direct readers to Lookahead's Figure 4 GIF, which we believe is far clearer than the paper itself.
>
> **On Lines 241 and 248.** We thank the reviewer for your careful read and they will be removed.
>
> ### **`W2 - No support for batched inference`**
> Upon benchmarking during development, we unfortunately cannot provide meaningful batched support, as some fundamental challenging limitations appear baked into the current n-gram paradigm: N-gram methods typically incur higher per-step FLOPs than vanilla decoding or SD methods (highlighted in Table 18), since they must perform both full-model verification and n-gram candidate generation within the same forward pass — making them far more likely to hit the compute-bound under batched settings. Such challenges suggest that a batch-capable n-gram design likely deserves a paper of its own.
>
> **We have faithfully highlighted this shortcoming with more details in the Limitations section of Appendix A.** In our defense, SD methods that support meaningful batching and those that do not essentially fall into different categories, often requiring distinct designs and targeting different deployment scenarios (e.g., TriForce vs. MagicDec, even though both originate from the same lab). This distinction becomes especially salient when large batch inference is involved. It is our understanding that draftless methods make the most sense for private, resource-constrained deployments — settings where batching demand is low but latency and task versatility are prioritized.
>
> ### **`W3 - compression ratio analysis`**
> We kindly refer the reviewer to Table 13 in our paper.
>
> ### **`Q1 - Novelty: what is the contribution beyond combination of exising works?`**
> We kindly refer the reviewer to our response to W1 of R`KRwJ`.
>
> ### **`Q2 -total FLOPs compared to vanilla decoding`**
> We kindly refer the reviewer to Table 18.
>
> ### **`Q3 - How to choose the optimal number of guesses?`**
> The optimal number of guesses reflects a trade-off between two competing effects: more guesses increase token acceptance rate (thus theoretical speedup), but also raise per-step FLOPs — eventually pushing the system into a compute-bound regime.
>
> This is empirically confirmed in our ablation studies (Tables 12 and 17). On A100, speedup improves from 1.05× (10 guesses) to 1.89× (20 guesses), but then degrades slightly to 1.75× at 30 guesses despite TFLOPs/step rising: a clear sign of compute saturation. In practice, **20–40 guesses** provides the best balance.
>
> In contrast, Lookahead Decoding peaks around 10 guesses and degrades sharply thereafter (Figure 2), as its full-KV draft generation becomes a bottleneck. FAFO's compressed-KV fumble decoding is precisely what allows it to scale to far more guesses without the same penalty, which is the core efficiency advantage FAFO holds over Lookahead.
>
> In summary, the optimal number of guesses is hardware- and task-dependent, but a simple grid search over {10, 20, 30, 40} on a representative task is sufficient, as the speedup curve is smooth and well-behaved. We will add this guidance to the updated version.

---

> > ### Author Rebuttal · Reviewer_5Qf5 · 2026-04-03
> >
> > Thanks for the rebuttal. It helps clarify some points.
> >
> > However my main concerns are still there. For presentation, shorter doesn’t mean less clear. Key ideas don’t need to be fully explained in the introduction. I still think the paper is not well structured.
> >
> > On novelty, it still feels like a combination of existing components rather than a clearly new contribution.
> >
> > So I’ll keep my score as weak reject.

---

> > > ### Author Response · Authors · 2026-04-04
> > >
> > > ### **`Key ideas don't need to be fully explained in the introduction. I still think the paper is not well structured.` We hear you, but having extended discussion is quite a tradition in self-SD/Lookahead literature. How about we meet in the middle and use separate Introduction and Background sections?**
> > >
> > > We thank the reviewer for the candid feedback. The presentation of FAFO is something we have struggled with quite a bit, and in our rebuttal we have clearly demonstrated that, even among the few reviewers of this paper, we have received very different preferences in this regard.
> > >
> > > For instance, your feedback indicates:
> > >
> > > > For presentation, shorter doesn't mean less clear. Key ideas don't need to be fully explained in the introduction.
> > >
> > > Which is in direct contrast to the feedback from `mHiU`, who liked the comparative discussion within the Introduction:
> > >
> > > > Clear presentation: The paper clearly contrasts FAFO with prior speculative decoding methods (TriForce, Lookahead, Self-SD).
> > >
> > > We also want to note that **our Section 1 is denoted as *Introduction and Background*, rather than *Introduction* alone.** Having relatively extended discussions is quite the norm in the n-gram decoding / self-speculative decoding literature, e.g.:
> > >
> > > * [TriForce](https://arxiv.org/pdf/2404.11912): 3 pages for Introduction.
> > > * [MagicDec](https://arxiv.org/pdf/2408.11049): 2.25 pages for Introduction.
> > > * [SparseSpec](https://arxiv.org/pdf/2512.01278): More than 2 pages of Introduction and around 1 page of Background.
> > > * [Lookahead Decoding](https://arxiv.org/pdf/2402.02057): 2.5 pages for Introduction and Background.
> > >
> > > In comparison, FAFO has a little over 3 pages for Introduction and Background, with a large visualization taking up half of Page 3. We therefore kindly argue that our presentation structure is well established in this line of literature.
> > >
> > > **That said, we take your feedback to heart and agree there is value in separating Introduction and Background into two sections.** Our content before Section 1.1 can in fact serve as a strong standalone Introduction. We will make a cut there, strengthen the new Introduction with additional detail from the bullet points at the end of Section 1.2, and turn Sections 1.1 and 1.2 into a standalone Background section — does that sound like a reasonable plan to you?
> > >
> > > ---
> > >
> > > ### **`It still feels like a combination of existing components rather than a clearly new contribution.` It is a combination of existing components, but that does not mean there is no meaningful contribution.**
> > >
> > > We kindly note that most, if not all, draftless + lossless efficient decoding methods are about identifying the right combination of existing techniques. For instance, self-speculative decoding methods integrate some form of KV cache compression with speculative decoding (StreamingLLM for TriForce, layer skipping for Swift, top-k attention for SparseSpec). Their contribution lies not in inventing every component from scratch, but in identifying a combination that enables a new capability regime.
> > >
> > > We argue FAFO does exactly that. Its contribution is not that each component is individually unprecedented, but that their integration yields a practically meaningful design with several benefits: a single KV cache for efficient deployment, task insensitivity for ease of use, strong performance for practical utility, and a cache manager for flexibility (which is particularly contributive, **as this is key to unlocking the training-free + lossless n-gram decoding paradigm that has been stagnant since Lookahead**). These are not cosmetic differences; they directly affect practical applicability.
> > >
> > > As one reviewer to another: **It is our genuine belief that it is easy to claim a method is just A + B and therefore lacks novelty**, and such criticism might well be generally true. What is harder, however, **is recognizing under what conditions there is an exception.** Otherwise, even works like Adapter Tuning / LoRA could be reduced to combining MLPs with selective finetuning — a rather boilerplate criticism.
> > >
> > > We rest our case here, and we hope this exchange invites some reflection from both sides on what constitutes a meaningful contribution to the community, regardless of whether the reviewer ultimately decides to revise their score.
> > >
> > > ---
> > >
> > > Update: It might be worth mentioning that our other reviewer, `KRwJ`, who previously raised a similar novelty/contribution concern:
> > >
> > > > Novelty/originality seems limited - much of the contribution looks like a systems-driven combination of existing ideas rather than a major new algorithm.
> > >
> > > Now finds our rebuttal fully resolves the concern:
> > >
> > > > **My concerns have been fully resolved, and I increase my score to a weak accept (4).**
> > >
> > > Though we respect your discretion in this regard.

---

### Official Review · Reviewer_BTdw · 2026-03-13

**Soundness:** 3
**Presentation:** 2
**Significance:** 3
**Originality:** 3
**Overall Recommendation:** 4
**Confidence:** 3

**Summary:**

This paper aims to leverage lossy KV cache compression strategies to accelerate decoding, while preserving lossless accuracy. They allow the model to use compressed KV cache to generate different n-gram options, and then verify these n-gram guesses in the same forward pass using lossless verification. They present custom system implementation to efficiently compute attention with these draft n-gram speculations. This method is compatible with different options for sparse attention, and provides substantial speedups relative to existing self-speculative decoding strategies without introducing additional memory footprint for the drafts.

**Compliance With Llm Reviewing Policy:**

Affirmed.

**Final Justification:**

Taking into account the rebuttal and discussion with other reviewers, I will maintain my positive assessment of the paper.

**Key Questions For Authors:**

- How does the proposed approach compare with using a trained n-gram model for generating a bunch of speculative continuations, and then verifying these in parallel?

**Limitations:**

Yes

**Strengths And Weaknesses:**

Strengths
- Their approach is compatible with different token pruning algorithms
- Their approach enables noticeable speedups without the overhead of typical speculative decoding methods (draft model, extra compressed KV entries, etc)
- Their system modifications enables generating many more drafts than prior methods for n-gram generation
- Their approach has strong latency results (which are also consistent across models/tasks)

Limitations:
- The formatting of the paper is hard to understand - it is not really self-contained (it relies on understanding prior lookahead decoding work, which is not well explained within the paper). I also felt that explaining the benefits of the method in section 3 before explaining the method in section 4 was confusing, and I had to jump around to understand the algorithm

---

> ### Author Rebuttal · Authors · 2026-03-31
>
> We thank the reviewer for the initial positive rating of the work, and we are glad that the reviewer finds our work strong. We hope to address your concerns below.
>
> ### **`W1 - Paper not self-contained; section 3 (benefits) before section 4 (method) is confusing`**
>
> **On writing and presentation:**
>
> We appreciate the specific and actionable feedback, and address each point below.
>
> **On self-containedness.** We sympathize with this concern, but want to be transparent:
> - Lookahead Decoding's original manuscript is, in our honest assessment, *not the clearest piece of writing*. Fully re-explaining it within our page budget would either crowd out FAFO's own content or produce a largely redundant rehash for readers already familiar with the paradigm.
> - We have instead tried to provide the most useful resources available: we strongly direct readers to Lookahead's Figure 4 GIF, which we believe is far clearer than the paper itself, and supplement this with our own Figure 1. We are happy to add a more thorough explanation of Lookahead in the appendix for the updated version.
>
>
> **On Section 3 appearing before Section 4.** The sequencing is intentional and we believe well-motivated, though we acknowledge it can feel disorienting if one expects a strict method-before-evaluation order. Section 3 is explicitly framed as a *practical overview and pilot study*, not an evaluation — designed to give readers an intuition for *why* FAFO's design choices are necessary before the full technical exposition in Section 4. This mirrors how many systems and efficiency papers present high-level motivation experiments before diving into implementation.
>
> That said, we take this feedback seriously. We would add a *clearer roadmap at the start of Section 3* explicitly signposting that Section 4 contains the full technical details, so readers know where to go for the algorithm and can treat Section 3 as optional motivating context rather than a prerequisite. We hope this reduces the need to "jump around."
>
>
> ### **`Q1 - How does FAFO compare to using a trained n-gram model for speculative continuations?`**
> We kindly refer the reviewer to our response to W2 of R`KRwJ`, where we provide a direct empirical comparison of FAFO against ANPD — a method that builds an adaptive multi-level n-gram module on-the-fly from the current context to generate speculative continuations, which are then verified in a single parallel forward pass. As shown in that table, FAFO-Stream consistently outperforms ANPD across all benchmarks and both models, despite ANPD being evaluated under favorable conditions (warm-up phase excluded).

---

> > ### Author Rebuttal · Reviewer_BTdw · 2026-04-04
> >
> > I appreciate the author's consideration of my feedback, particularly around writing style. I think adding a more thorough explanation in the appendix (rather than a third-party resource) would be ideal. I think that a roadmap-style introduction at the start of section 3 would also be valuable.
> >
> > The response to Q1 address my other question. I will retain my positive assessment of the paper.

---

### Official Review · Reviewer_mHiU · 2026-03-13

**Soundness:** 3
**Presentation:** 3
**Significance:** 2
**Originality:** 2
**Overall Recommendation:** 4
**Confidence:** 5

**Summary:**

The paper proposes FAFO (Fumble Around, Find Out), a draftless decoding framework that integrates lossy KV-cache compression with lossless verification within a single forward pass.
Rather than relying on a separate draft model, FAFO generates n-gram candidates (typically 2-grams) from a compressed KV cache (“Fumble Around”) and verifies them against the full KV cache (“Find Out”) in parallel, using a custom sparse attention mask built on FlexAttention.
The authors report 1.20–2.71x latency speedups across multiple models (Llama-2/3, Qwen2.5) and benchmarks while preserving lossless generation quality.

**Compliance With Llm Reviewing Policy:**

Affirmed.

**Final Justification:**

My concerns are adequately addressed. I will maintain my positive recommendation.

**Key Questions For Authors:**

+ Have authors try using FlashInfer's Kernel [1] for supporting the mixed sparse and dense attention with page size equal to 1 (i.e., block size)? It seems like the FAFO cache management strategy stems from the block size limitation of FlexAttention due to its Triton backend.
+ Can the author add a comparison to the latest self-spec methods, such as SparseSpec [2]?

[1] https://github.com/flashinfer-ai/flashinfer/blob/f487726405e173be29a26a3371dcb68a96243154/flashinfer/attention.py#L43
[2] Accelerating Large-Scale Reasoning Model Inference with Sparse Self-Speculative Decoding

**Limitations:**

yes

**Strengths And Weaknesses:**

**Strengths**

+ Interesting idea: FAFO’s capability to perform n-gram drafting using a compressed KV cache and verify the n-gram drafts within the same forward pass is novel and technically elegant.

+ Clear presentation: The paper clearly contrasts FAFO with prior speculative decoding methods (TriForce, Lookahead, Self-SD). Figure 2 effectively illustrates the mechanism and makes the concept easy to grasp.

**Weaknesses**
+ Limited workload generalizability. The FAFO only supports batch size 1. While conservativly defense in the limitation.

---

> ### Author Rebuttal · Authors · 2026-03-31
>
> We thank the reviewer for the initial positive rating of the work, and we are glad that the reviewer finds our work interesting. We hope to address your concerns below.
>
> ### **`Q1 - FlashInfer's kernel for mixed sparse/dense attention`**
>
> We use FlashInfer's single-sequence prefill kernel with a precomputed, token-level boolean attention mask that encodes the exact FAFO access pattern (sink tokens + sliding window + speculative frame). This achieves token-level mask granularity without the fixed block-size constraint of FlexAttention's Triton backend, and without any paging overhead.
>
> We present results on Llama-3.1-8B-Instruct with the same setting as we reported in the paper.
>
> | Method | MT-Bench Speedup | MT-Bench τ | GSM8K Speedup | GSM8K τ | HumanEval Speedup | HumanEval τ |
> |---|---|---|---|---|---|---|
> | FAFO-FlexAttention | 1.50× | 2.10 | 1.31× | 2.08 | 1.57× | 2.20 |
> | FAFO-FlashInfer | 1.45× | 2.54 | 2.07× | 3.02 | 2.10× | 2.31 |
>
> Interestingly, while FAFO-FlashInfer achieves substantially higher speedup on GSM8K and HumanEval, it slightly underperforms on MT-Bench. We attribute this to task characteristics: mathematical reasoning and code generation exhibit highly structured, repetitive token patterns that yield strong n-gram speculation candidates, whereas open-ended multi-turn dialogue is more lexically diverse — reducing the density of reusable n-gram chains and leading to a modest reduction in speculative acceptance rate. The higher τ is consistent with the observation in Appendix H as different attention kernels may produce slightly different decoded outputs, which can affect n-gram generation and verification process.
>
>
> ### **`Q2 - Comparison to SparseSpec`**
>
> We thank the reviewer for raising this comparison. We note that SparseSpec was released on December 1st, less than two months before the ICML submission deadline, which makes *a head-to-head comparison arguably outside the scope of contemporaneous work per ICML guidelines*. Nonetheless, we provide a full comparison below in the spirit of completeness.
>
> As shown in the table, SparseSpec achieves higher raw speedups on most benchmarks. However, we argue that this gap is primarily attributable to **system-level engineering advantages** that SparseSpec has accumulated: CUDA graph capture with double buffering, fused draft+verify FlashInfer kernels, asynchronous CPU scheduling, and KV cache offloading.
>
> To validate this, we integrated FlashInfer kernel support into FAFO per the suggestion of the reviewer. As shown in the table, FAFO (FlashInfer) substantially closes the gap on GSM8K (2.07× vs. 2.15×) and HumanEval (2.10× vs. 2.73×), with only a simple system-level support added, unlike carefully designed system like SparseSpec. This confirms that the remaining throughput difference reflects ongoing system engineering work orthogonal to the core research contribution, and we view full system optimization as complementary future work.
>
> | Method | GSM8K Speedup | HumanEval Speedup | MT-Bench Speedup |
> |--------|--------------|-------------------|-----------------|
> | SparseSpec | 2.15× | 2.73× | 2.47× |
> | FAFO (FlexAttention) | 1.31× | 1.57× | 1.50× |
> | FAFO (FlashInfer) | 2.07× | 2.10× | 1.45× |

---

> > ### Author Rebuttal · Reviewer_mHiU · 2026-04-03
> >
> > Thanks for the Author rebutal. Reviewers understand that SparseSpec is a relatively new work, but to make FAFO more comprehensive and robust, they still recommend that the author discuss and compare it with SparseSpec, given their potential similarities in the questions FAFO is working on and the strategies they adopt.
> >
> > Aside from this, can authors provide a more detailed discussion on the comparison with respect to the SparseSpec? As far as reviewers know, SparseSpec also includes a new sparse attention technique for producing draft tokens.  Can FAFO be combined with SparseSpec's pillar attention to make n-gram pools more informative?
> >
> > Also, I would like to follow up with another question. Can the author provide more details on maintaining the n-gram pool? Does 'n' have a fixed constant? Has the author balanced across different 'n'?

---

> > > ### Author Response · Authors · 2026-04-04
> > >
> > > ### **`Follow-up #1 - Comparison with SparseSpec` Yes, we will cite & discuss SparseSpec, here's a draft**
> > >
> > > We agree with the reviewer that SparseSpec should be properly discussed, especially given it is likely the strongest self-speculative decoding (SSD) method to date.
> > >
> > > Our work has spent a fair amount of real estate discussing many existing SSD methods, and SparseSpec should certainly be included. Our mention of SparseSpec being within the two-month concurrent work window is more of a preemptive defense of performance (especially because SparseSpec is a much more system-heavy work with impressive throughput), rather than a rejection to discuss it. Here, we provide a sketch of the comparative discussion:
> > >
> > > The strongest similarity between FAFO and SparseSpec is that they are both draftless + lossless efficient decoding methods. Beyond that, both methods employ KV cache compression as part of their backbone. More specifically:
> > >
> > > **SSD vs n-gram** SparseSpec follows *sequential* speculative decoding: it runs `spec_stride=8` autoregressive draft steps with a *specific* compressed KV cache, then a single verification forward pass, following the typical draft-*then*-verify design for SSD methods. FAFO, by contrast, is a *parallel* n-gram candidate pool decoder: draft candidates are generated from the pool on-the-fly, and both drafting and verification execute within a *single* forward pass — essentially draft-*and*-verify simultaneously. This distinction is the same one that separates FAFO from all SSD methods (Section 2.2 of the paper).
> > >
> > > **On the sparse attention technique.** SparseSpec's Pillar attention is essentially a KV cache compression design: it reuses the attention scores computed during verification to identify the top-K salient tokens at no additional cost, then carries that sparsity pattern into the next `spec_stride` draft steps. FAFO does not propose a new compression kernel; instead, it is designed as a framework that aims to be *agnostic* to the underlying compression method — StreamingLLM, Quest, or a method like Pillar can all serve as FAFO's "fumble" backend.
> > >
> > > Please do let us know if there is anything else you'd prefer us to highlight regarding SparseSpec.
> > >
> > > ---
> > >
> > > ### **`Follow-up #2 - FAFO's compatibility with PillarAttention` Yes!**
> > >
> > > PillarAttention periodically selects the top-K most important KV pages by scoring all pool tokens against the verification queries, keeping only the highest-attention pages in the attended region. Much like Quest, PillarAttention is integrable into FAFO's fumble decoding.
> > >
> > > We preliminarily integrated PillarAttention as a KV compression method in FAFO and evaluated it on Llama-3.1:
> > >
> > > | Benchmark | Speedup vs Dense | Compression Ratio |
> > > |---|---|---|
> > > | MT-Bench | 1.18× | 2.58× |
> > > | HumanEval | 1.11× | 2.65× |
> > >
> > > The speedups are moderate because our implementation has to re-materialize attention scores after each verification pass to identify top-K pages, whereas SparseSpec avoids this overhead through a custom fused CUDA kernel. We did not implement such a kernel given the limited time of rebuttal, but we believe it could further improve throughput.
> > >
> > >
> > > ---
> > > ### **`Follow-up #3 - n-gram pool details`**
> > >
> > > We thank the reviewer for this question. In short, $n$ is an adjustable hyperparameter, but it is a constant once set.
> > >
> > > Details on pool management are provided in **Appendix E**. The n-gram pool stores candidate subsequences indexed by their starting token. The pool size per starting token is capped at a fixed maximum, with **LRU eviction** to discard the least recently used entries when the pool is full. This ensures bounded memory usage while retaining the most recently relevant candidates.
> > >
> > > Regarding the value of *n*: yes, *n* is a fixed constant set at runtime. We conduct an ablation over *n* ∈ {4, 5, 6, 7, 8, 9} on an A100 GPU in **Appendix F.7 (Table 11)**. The results show an optimum around *n* = 6–7, achieving ~1.90× speedup. Shorter n-grams (*n* = 4) perform poorly because they are too short to meaningfully amortize the verification overhead. Longer n-grams (*n* = 8–9) also underperform because matching a longer exact subsequence is rarer, reducing acceptance rates, while also spending more computational resources to verify the longer candidate sequences.
> > >
> > > (One editorial nuance is we use $k$ for the length of the n-gram, but $n$ for the number of n-gram guesses when declaring/ablation hyperparameters — just a heads up.)

---

### Official Review · Reviewer_KRwJ · 2026-03-13

**Soundness:** 3
**Presentation:** 2
**Significance:** 3
**Originality:** 2
**Overall Recommendation:** 4
**Confidence:** 4

**Summary:**

FAFO uses lossy token-dropping KV compression to generate many speculative n-gram guesses, stores them in a prefix-indexed candidate pool, and verifies them losslessly in the same forward pass with a custom FlexAttention-based KV manager.

**Compliance With Llm Reviewing Policy:**

Affirmed.

**Final Justification:**

The rebuttal addressed my concerns well.

**Key Questions For Authors:**

- Can you compare directly against stronger recent baselines such as LongSpec, QuantSpec, and TokenSwift
- How exactly does FAFO differ algorithmically from prior exact n-gram/parallel decoding methods beyond Lookahead, especially ANPD?

If the novelty issues are sufficiently addressed and comparison to recent baselines are included (with strong results), I will increase my score.

**Limitations:**

yes

**Strengths And Weaknesses:**

Strengths:
- Strong motivation and important problem area.
- Strong practical systems integration: lossy KV “fumble” generation, n-gram candidate-pool decoding, and same-pass full-cache verification are combined in a coherent way.
- Proposed architecture is good for memory-constrained settings.
- Strong breadth of baselines for experiments.
- The reported speedups are meaningful enough to make the method practically interesting if the exactness and deployment assumptions hold.

Weaknesses:
- Novelty/originality seems limited - much of the contribution looks like a systems-driven combination of existing ideas rather than a major new algorithm.
- Related-work positioning is incomplete relative to recent speculative/long-context decoding literature. The paper does not discuss ANPD [1], LongSpec [2], QuantSpec [3], and TokenSwift [4].
- The theoretical support is weak; the main proposition is a fairly limited monotonicity result and does not strongly justify the retrieval strategy.
- As noted above regarding related work, more recent baselines are missing. Additionally, some appendix benchmarks do not provide direct head-to-head comparisons.

[1] https://arxiv.org/abs/2404.08698v1

[2] https://arxiv.org/abs/2502.17421

[3] https://arxiv.org/abs/2502.10424

[4] https://arxiv.org/abs/2502.18890

---

> ### Author Rebuttal · Authors · 2026-03-31
>
> ### **`W1, Q2 - Limited novelty`**
>
> We agree that our work largely adopts lossy KV cache compression under the n-gram context and lacks major algorithmic novelty. However, algorithmic novelty is only one part of a contribution, where the context of a work's respective landscape matters — please hear us out though some hypothetical Q&As:
>
> - **`Are there must-have constraints limiting the freedom of algorithmic novelty?` Yes — training-free + draftless + lossless decoding is highly practical but also extremely restrictive.**
> 	- Common “novelty avenues” — custom training, architecture changes, draft/target co-design, lossy trade-offs — are off the table.
> 	- Prior art under these constraints also reuses mature components: TriForce (COLM 2024), MagicDec (ICLR 2025), and SWIFT (ICLR 2025) all rely on established KV compression with speculative decoding.
> - **`Is the field saturated with random repackaging?` No — there has been no lossless + draftless successor to Lookahead since its late-2023 debut.**
> 	- If many lossy KV + n-gram methods existed, FAFO might look like random swapping and be of little value. But there has been zero lossless follow-up to Lookahead in two years, despite many lossy KV + SD proposals being proposed in the same period.
> 	- This suggests randomness doesn’t work — real progress needs deep understanding, careful component choice, and meticulous execution. We venture to argue FAFO delivers exactly that.
> - **`Is the performance gain massive?` Yes — FAFO’s speedup is exceptional.**
> 	- The efficiency community values simplicity with impact. FAFO fixes Lookahead’s key drawback (unable to host more n-gram guesses) and delivers unmatched end-to-end acceleration under hard constraints across a broad task suite.
> - **`Rerouting known ingredients to solve a harder problem is a valid contribution.` FAFO meaningfully revives the stagnated n-gram candidate-decoding research space.**
> 	- As the risk of being massively redundant, we highlight that many impactful efficiency works are algorithmically light but component-savvy:
> 		- QLoRA = standard quantization + LoRA
> 		- KIVI = per-channel quantization + sliding-window buffer
> 		- MEND = gradient decomposition + editing
> 	- We also believe the community would benefit more from having a FlexAttention interface enabling many lossy KV methods, than from us hardcoding a single unique method purely for scoring more novelty points.
>
> We are firm believers in one ARR guideline:
>
> > *“**The goal is to solve the problem, not to solve it in a complex way.** Simpler solutions are preferable — they are less brittle and easier to deploy.”*
>
> And we hope the reviewer shares this philosophy.
>
> ### **`W2, W4, Q1 - Missing baseline comparisons`**
> We note that LongSpec, and TokenSwift are briefly mentioned in our *extended related work* section. Regarding direct comparison: we include SWIFT in our paper, which its authors explicitly benchmark against TokenSwift and demonstrate superior performance. We therefore treat SWIFT as a strong proxy for TokenSwift.
>
> To further address this concern, we ran ANPD using its released code and compare it alongside SWIFT and FAFO. Note that we report ANPD throughput excluding its n-gram warm-up phase, giving it the most favorable measurement:
>
> **Llama-3.1-8B-Instruct**
>
> | Method | MT-Bench Speedup | τ | GSM8K Speedup | τ | HumanEval Speedup | τ |
> |---|---|---|---|---|---|---|
> | FAFO-Stream | 1.50× | 2.10 | 1.31× | 2.08 | 1.57× | 2.20 |
> | SWIFT | 0.94× | 2.93 | 0.98× | 3.22 | 1.03× | 2.38 |
> | ANPD | 0.87× | 1.53 | 0.52× | 1.52 | 0.78× | 1.60 |
>
> FAFO-Stream outperforms both ANPD and SWIFT across all benchmarks and both models.
>
>
> **FAFO differs fundamentally from ANPD:** Although both methods use n-gram candidates without a draft model, they diverge significantly in design. ANPD retrieves candidates from a fixed n-gram database built during a warm-up phase. FAFO instead maintains a *dynamic* token pool continuously updated from the ongoing context, enabling it to adapt to the current generation.
>
> **On incomplete head-to-head comparisons in the appendix:** Some full head-to-head comparisons are constrained by baseline compatibility, not by omission. For example, TriForce uses a non-standard draft model (JF68M) with no equivalent for other model families such as Qwen — making a fair comparison structurally impossible, not merely inconvenient. We thank the reviewer for this suggestion and will fill in all feasible numbers in the appendix in the updated version.
>
> ### **`W3 - Theoretical support weak`**
>
> We thank the reviewer for this observation, but wish to clarify intent: we never claim theoretical novelty as a contribution of FAFO. The monotonicity result is included solely to provide intuition for *why* the retrieval strategy works, offering a principled explanation for an empirically effective design choice, not a theoretical foundation for the method. We will make this framing more explicit in the updated version to avoid overclaiming.

---

> > ### Author Rebuttal · Reviewer_KRwJ · 2026-04-04
> >
> > My concerns have been fully resolved, and I increase my score to a weak accept (4).

---

> > > ### Author Response · Authors · 2026-04-07
> > >
> > > We thank this reviewer for finding that our rebuttal fully resolved your concerns, especially on the novelty/contribution end. We just want to kindly remind you that you have not yet increased our rating to weak accept (4), despite your writing.
> > >
> > > If you could be so kind as to reflect that in the system, we would appreciate it.
> > >
> > > *Paper19106 Authors*

---

### Decision · Program_Chairs · 2026-04-30

**Decision:**

Accept (regular)

**Comment:**

This paper presents a new method for speeding up LLM text generation, while preserving output quality. To achieve this, they use a compressed KV cache to generate candidate continuations. Then they verify all the (short) candidate continuations in parallel against the full KV cache in a single forward pass.

The reviewers considered this method technically sound and practically relevant. They appreciated the strong latency results and the breadth of evaluation. The rebuttal successfully addressed several concerns, regarding comparison to more recent baselines, and clarified implementation choices.

One reviewer remained unconvinced about the novely of the work, viewing it as a mix of existing techniques, and felt like the paper's structure and clarity could be improved. One major limitation of the paper stems from the fact that the proposed method can only handle batches of size 1 (which the authors clearly acknowledge), and it is less clear what kind of speedups this would allow when trying to serve many prompts in parallel.

Despite these weaknesses, this can be viewed as a well executed systems paper which offers a meaningful practical contribution in an important area.